# The ELAV/Hu protein Found in neurons regulates cytoskeletal and ECM adhesion inputs for space-filling dendrite growth

Rebecca A. Alizzi[1], Derek Xu[1], Conrad M. Tenenbaum[1], Wei Wang[2], Elizabeth R. Gavis[1]*

1 Department of Molecular Biology, Princeton University, Princeton, New Jersey, United States of America, 2 Lewis-Sigler Institute, Princeton University, Princeton, New Jersey, United States of America

* gavis@princeton.edu

**Data Availability Statement:** All data except for the TRIBE analysis raw sequencing data are provided in the manuscript and in its Supporting information files. The TRIBE raw sequencing data and the

## Abstract

Dendritic arbor morphology influences how neurons receive and integrate extracellular signals. We show that the ELAV/Hu family RNA-binding protein Found in neurons (Fne) is required for space-filling dendrite growth to generate highly branched arbors of *Drosophila* larval class IV dendritic arborization neurons. Dendrites of *fne* mutant neurons are shorter and more dynamic than in wild-type, leading to decreased arbor coverage. These defects result from both a decrease in stable microtubules and loss of dendrite-substrate interactions within the arbor. Identification of transcripts encoding cytoskeletal regulators and cell-cell and cell-ECM interacting proteins as Fne targets using TRIBE further supports these results. Analysis of one target, encoding the cell adhesion protein Basigin, indicates that the cytoskeletal defects contributing to branch instability in *fne* mutant neurons are due in part to decreased Basigin expression. The ability of Fne to coordinately regulate the cytoskeleton and dendrite-substrate interactions in neurons may shed light on the behavior of cancer cells ectopically expressing ELAV/Hu proteins.

## Author summary

Different types of neurons have different sizes and shapes that are tailored to their particular functions. In the fruit fly larva, a set of sensory neurons called class IV da neurons have highly branched trees of dendrites that cover the epidermis to sense potentially harmful stimuli. Neurons whose dendrites completely fill the territory they sample are also found in zebrafish, worms, mice and humans. We show that an RNA-binding protein called Fne plays an important role in coordinating different contributions to dendrite branching in class IV da neurons by impacting the organization of the cytoskeleton within the neuron and the ability of the dendrite to contact the substratum outside of it. The identification of mRNAs that code for cytoskeleton regulators and adhesive proteins as targets of Fne using a genome-wide approach further supports these results. While the ability of Fne to exert control over such proteins is crucial to generating the space-filling growth of

processed RNA editing tracks have been deposited in NCBI's Gene Expression Omnibus and are accessible through GEO Series accession number GSE160111 (https://www.ncbi.nlm.nih.gov/geo/query/acc.cgi?acc=GSE160111).

**Funding:** This work was supported by National Institute of Health (NIH) grants R01 GM067758 and R35 GM126967 to ERG. RAA and CMT were supported by NIH training grant T32 GM007388. The funders had no role in study design, data collection and analysis, decision to publish, or preparation of the manuscript.

**Competing interests:** The authors have declared that no competing interests exist.

neurons, it can be deleterious if not properly employed, such as when the homologs of Fne are expressed in cancer cells.

## Introduction

Neurons display a diversity of dendrite morphologies that are tailored to their particular synaptic or sensory functions. The size, shape, and complexity of the dendritic arbor are the result of developmental programs that regulate dendrite outgrowth and branching to establish, refine, and then maintain coverage of the receptive field. Cell-intrinsic mechanisms as well as interactions between the neuron and the surrounding environment drive patterning of the arbor [1–3] but how these processes are controlled and coordinated to generate specific dendritic branching patterns remains incompletely understood.

The *Drosophila* larval dendritic arborization (da) neurons are a well-established model system for studying dendrite patterning. These sensory neurons comprise four morphologically and functionally distinct classes with overlapping territories along the larval body wall [4]. Among them, the class IV da neurons have the most complex dendritic arbors and respond to noxious stimuli [5]. Class IV da neurons elaborate highly branched, two-dimensional dendritic arbors between the epidermis and the underlying extracellular matrix (ECM) that completely and non-redundantly tile the larval body wall [4,6,7]. The uniform, space-filling morphology of these arbors is the result of a highly dynamic growth and refinement process as well as repulsive interactions between neighboring dendrites [1,2]. After the tiled pattern of class IV da neurons is established early in larval development, the dendrites remain plastic. New branches extend and retract, allowing the arbor to expand in proportion to the larval body wall as it grows dramatically over the next several days [8]. As larval growth reaches its endpoint, dendritic branches must stabilize to maintain proper innervation of the epidermis [9].

The execution of dendrite branching relies on cell biological processes integral to the neuron including membrane trafficking and cytoskeletal dynamics [1,2]. The characteristic architecture of class IV da neuron arbors is genetically controlled by transcription factors that directly or indirectly regulate effectors of actin and microtubule organization as well as components of the secretory machinery [10–16]. In class IV da neurons, the deployment of these neuron-intrinsic factors in dendrite patterning is influenced by external inputs including integrin-mediated interactions between the dendrites and underlying ECM and interactions between the dendrites and epidermis [6,7,17–20]. We now provide evidence that control of cytoskeletal composition and neuronal-substrate interactions is coordinated post-transcriptionally by an RNA-binding protein (RBP), the Elav/Hu family protein Found in neurons (Fne).

The Elav/Hu protein family is a group of conserved RBPs that function at multiple steps in RNA metabolism, including alternative splicing, alternative polyadenylation, localization, translation, and transcript stabilization [21,22]. In humans, this family comprises three neuronal proteins (HuB, HuC, and HuD) and one ubiquitous protein (HuR) [21]. Neuronal Hu proteins play important roles in neuron differentiation, dendrite and axon outgrowth, and plasticity through their regulation of numerous transcripts [22,23]. In addition, both neuronal and ubiquitous Hu proteins are expressed in certain types of cancer cells [24,25]. In *Drosophila*, the Elav/Hu protein family consists of three members: Elav, Fne and Rbp9. *fne* expression is restricted to neurons [26,27] and, in contrast to Elav and Rbp9, Fne is primarily cytoplasmic [26]. Previous work has identified roles for Fne in regulating synaptic plasticity, mushroom body development, and male courtship behavior [28,29]. Fne, Elav, and HuD have all been

implicated in dendrite morphogenesis [30,31], but what roles these proteins play in this process and the relevant RNA targets remain unknown.

Here, we show that Fne is required in class IV da neurons throughout larval development to promote the space-filling growth by which these neurons establish and maintain coverage of their receptive fields. Genetic analysis revealed Fne regulates targets that participate in cytoskeletal organization and dendrite-ECM interactions. Consistent with this role, targets of Fne identified by TRIBE (Targets of RNA-binding proteins Identified By Editing) were enriched for transcripts encoding cell adhesion proteins and cytoskeletal regulators. The TRIBE results suggest that the morphological defects seen in *fne* mutant neurons are the result of misregulation of many target transcripts. Analysis of one target, which encodes the cell adhesion protein Basigin (Bsg), showed that Fne regulation of *bsg* accounts in part for the cellular characteristics of these neurons required for space-filling dendritic growth. Our results reveal a role for Fne in coordinating the cytoskeletal organization and dendrite-substrate interactions necessary to modulate branching dynamics and promote stable dendrite growth.

## Results

### Fne is required throughout larval development for the space-filling arborization of class IV da neurons

We identified Fne in an RNAi screen for RBPs required in class IV da neurons for dendrite morphogenesis [30]. Anti-Fne immunofluorescence revealed Fne expression in the soma of class IV da neurons through the third larval stage, with levels decreasing dramatically during pupariation (S1 Fig). Because the screen only detected defects in these neurons at the endpoint of arbor morphogenesis, we performed a developmental analysis of *fne* null mutant (*fne$^-$*) neurons to further investigate the role of Fne. The initial territories of class IV da neurons are established by the end of embryogenesis. At this time, *fne$^-$* and wild-type arbors appeared qualitatively similar and there were no significant differences in terminal branch number, terminal branch length or total dendrite length (Fig 1A, 1B and 1J–1L). Development then proceeds through three larval stages (L1, L2, and L3) during which the animal's surface area increases ~100 fold [32]. By the end of L1, wild-type arbors fill the surrounding territory and develop the space-filling morphology characteristic of class IV da neurons. At this stage, *fne$^-$* arbors had produced an excess of short terminal branches as compared to wild-type (1.8-fold, $p < 0.01$), resulting in increased coverage of the dendritic field (empty space = 43% in WT, 37% in *fne$^-$*, $p < 0.05$) (Fig 1C, 1D and 1J–1M). Wild-type neurons continued to arborize during L2, leading to an increase in field coverage as development progressed. *fne$^-$* neurons still showed an excess of terminal branches at this time point (1.8-fold, $p < 0.001$), but these short branches no longer provided additional coverage relative to wild-type neurons as the field expanded (empty space = 29% in WT, 30% in *fne$^-$*) (Fig 1E, 1F and 1J–1M). Terminal branches in *fne$^-$* neurons remain shorter than in wild-type through the end of L3 (Fig 1G, 1H and 1K). However, Sholl analysis revealed a decrease in branches close to the soma at L3 in *fne$^-$* neurons as compared to wild-type neurons (Fig 1N and 1O), suggesting that the proximal branches that had been abundant in L2 were lost by L3. Finally, in contrast to wild-type neurons whose branches rarely cross each other, *fne$^-$* neurons showed a 2.4-fold increase in isoneuronal dendrite crossing events ($p < 0.001$) (Fig 1P–1R). Together with the reduction in terminal branch length, this rearrangement in the architecture of the arbor resulted in a dramatic loss of field coverage (empty space = 16% in WT, 28% in *fne$^-$*, $p < 0.001$) (Fig 1M).

Because dendrite patterning is influenced by the overlying epidermal cells, we confirmed that Fne function is neuron-specific using several tests. First, re-expression of *fne* in *fne$^-$* neurons rescued the field coverage defect (S2A–S2D Fig). Second, expression of either of two

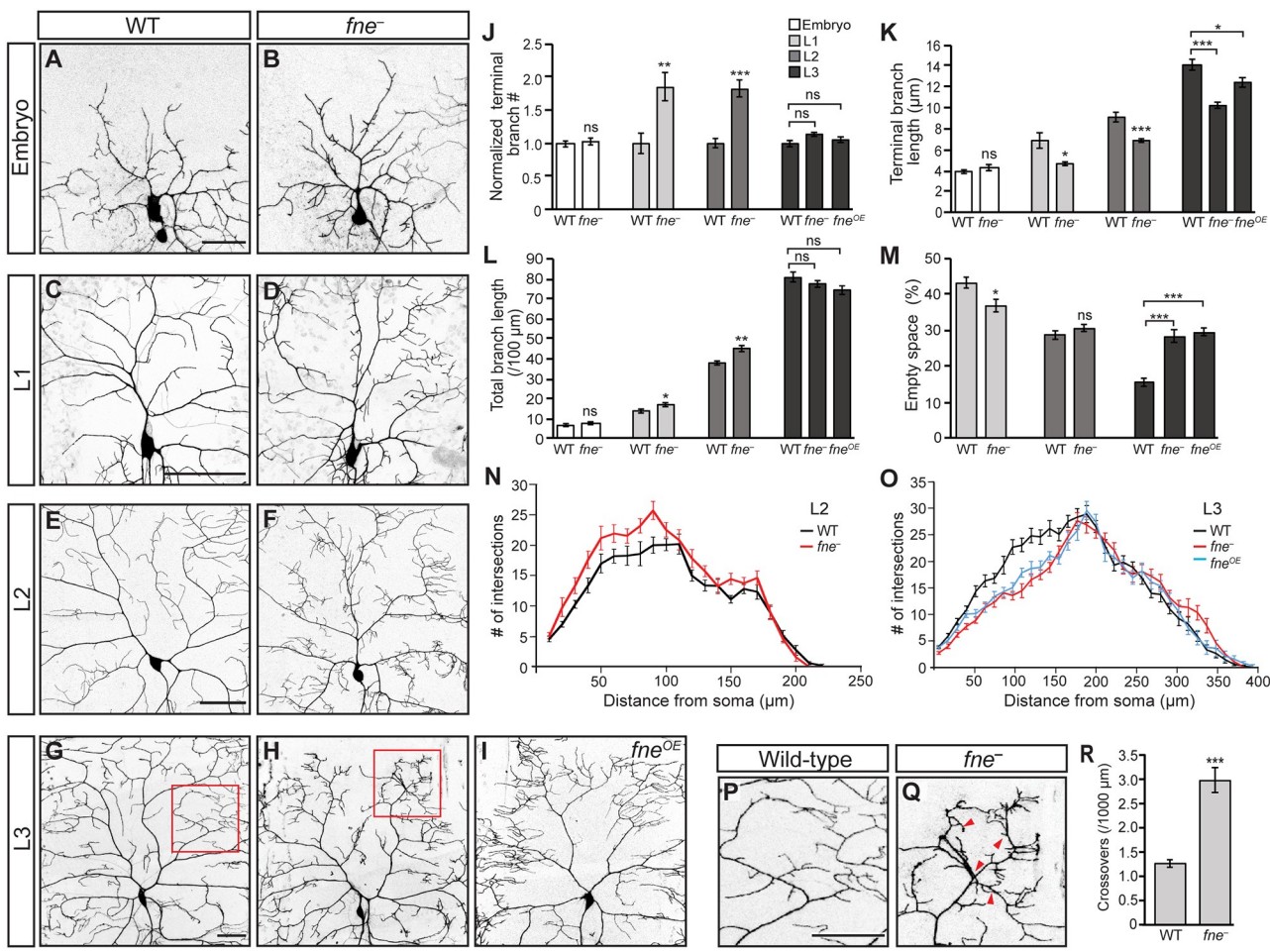

**Fig 1. Fne is required beginning in L1 for the space-filling arborization of class IV da neurons.** (A-I) Confocal z-series projections of representative class IV da neurons from wild-type (WT; A, C, E, G), *fne⁻*(B, D, F, H), or *fne^{OE}* (I) larvae at the end of embryogenesis (A, B), L1 (C, D), L2 (E, F), and L3 (G–I). For *fne^{OE}*, *ppk-GAL4* was used to drive expression of *UAS-fne*. For all genotypes, *ppk-GAL4* was used to express *UAS-CD4-tdGFP* to mark the neurons. (J) Quantification of terminal branch number. The values for the *fne⁻* and *fne^{OE}* neurons are normalized to the number of terminal branches in wild-type neurons at each time point. (K) Quantification of terminal branch length. (L) Quantification of total dendrite length. (M) Quantification of the amount of empty space within the arbor. (N, O) Sholl analysis of the indicated genotypes at L2 (N) and L3 (O). (P, Q) Magnified view of the areas outlined in red in G and H, with red arrows indicating points of dendrite crossing over events. (R) Quantification of all crossover events in WT and *fne⁻* neurons normalized to branch length. For (J-O, R) n = 10 neurons (WT, embryo), 10 neurons (*fne⁻*, embryo), 6 neurons (WT, L1), 6 neurons (*fne⁻*, L1), 10 neurons (WT, L2), 11 neurons (*fne⁻*, L2), 13 neurons (WT, L3), 10 neurons (*fne⁻*, L3), 17 neurons (*fne^{OE}*, L3). Two-tailed Student's *t*-test was used to determine significance between WT and *fne⁻* at the embryo, L1 and L2 stages (J-M) and in (R). The one-way ANOVA with Bonferroni-Holm *post hoc* test (J, K, M) or the Kruskal-Wallis with Dunn's *post hoc* test (L) was used to determine significance for L3. Values are mean ± s.e.m.; ns = not significant, *p<0.05, **p<0.01, ***p<0.001. Scale bars: (A) 20 μm, (B-D, P) 50 μm.

independent *UAS-fneRNAi* lines using a class IV da neuron-specific GAL4 driver produced arbor coverage and dendrite crossing defects similar to those of *fne⁻* neurons (S2E–S2G, S2E'–S2G' and S2H Fig). Finally, expression of *UAS-fneRNAi* in the epidermis had no effect (S2I–S2N Fig). Together, these results demonstrate a cell-autonomous function for Fne in class IV da neurons.

We also examined the effects of excess Fne by using *ppk-GAL4* to drive expression of *UAS-fne* (*fne^{OE}*) in wild-type class IV da neurons. Defects in *fne^{OE}* neurons were first evident during L3, with a decrease in terminal branch length (WT = 14 μm, *fne^{OE}* = 12 μm, p<0.05) (Fig 1I and 1K). Arbors of *fne^{OE}* neurons showed reduced branching proximal to the soma as compared to wild-type arbors (Fig 1O) and the higher order branches appeared clustered together

(Fig 1I). Similarly to $fne^-$ neurons, the reduction in terminal branch length combined with the rearrangement in the arbor architecture resulted in loss of field coverage (empty space = 16% in WT, 29% in $fne^{OE}$, p<0.001) (Fig 1M).

Defects caused by the loss or excess of *fne* were also apparent in the pupal stage. During pupariation, the dendrites of class IV da neurons are completely pruned and subsequently regrow to innervate the adult epidermis. Pruning occurs in a stereotypical time sequence that includes severing of the main dendrite branches followed by fragmentation and clearance [33,34]. In wild-type pupae, class IV da neurons exhibited thinning and severing of main branches by 7 h after puparium formation (APF), but the arbor remained largely intact. By contrast, in $fne^-$ neurons, the majority of the arbor was cleared by this time (15 branches remaining in WT, 3.7 branches in $fne^-$, p<0.001). Conversely, pruning was delayed in $fne^{OE}$ neurons. Whereas arbors in wild-type neurons were typically cleared by 16 h APF, $fne^{OE}$ arbors remained largely intact (0 branches remaining in WT, 14 branches in $fne^{OE}$, p<0.001) (S3 Fig). These results are consistent with the dramatic drop in Fne levels during pupariation (S1 Fig) and suggest that Fne normally acts as a brake, delaying the onset of pruning in class IV da neurons until the appropriate developmental time.

In summary, results from this temporal analysis show that *fne* is required in class IV da neurons for dendrite morphogenesis from L1 until pruning during pupariation. Whereas Fne is not required for branching per se, it functions in arbor patterning to promote the space-filling morphology characteristic of class IV da neurons.

## Fne is required for stable branch growth

In both *fne* mutant and overexpressing neurons, the transition from L2 to L3 coincides with the appearance of field coverage defects. To determine how these defects arise, we monitored dendrite branching in individual neurons over time by live imaging. Between the end of L2 and the end of L3, wild-type neurons produced an average of 125 new branches. In contrast, $fne^-$ neurons produced an average of 238 new branches during the same time period (p<0.01) (Fig 2A–2D and 2G). Furthermore, whereas 29% of existing branches were eliminated in wild-type neurons, 63% of branches were eliminated in $fne^-$ neurons (p<0.01) (Fig 2H). These data suggest that branches in *fne* mutant arbors are highly dynamic and less stable than in wild-type arbors. Although some $fne^{OE}$ neurons showed bursts of terminal branches sprouting as larvae progressed through development, this was not consistent among neurons (Fig 2E–2H).

The decreased branch stability in *fne* mutant neurons was also observed on a shorter time-scale. Time-lapse imaging over a period of 12 minutes during late L3 showed that the proportion of dynamic branches in $fne^-$ neurons was 2.6-fold greater than that of wild-type neurons (p<0.01) (Fig 2I–2K). Furthermore, some terminal branches in $fne^-$ neurons underwent multiple growth and retraction events during this period, whereas wild-type branches remained largely static. These behaviors suggest that Fne promotes stable dendrite outgrowth, ultimately generating the space-filling morphology characteristic of class IV da neurons.

## Microtubule content of the arbor is affected by Fne

Many of the intrinsic and extrinsic inputs to dendrite morphogenesis ultimately impact cytoskeletal organization and dynamics. In class IV da neurons, filamentous actin is distributed intermittently throughout the arbor [12,14,35], and remodeling of actin by Arp2/3 followed by actin polymerization initiates branch formation along primary dendrites [35,36]. Microtubule content also plays a key role in branching pattern and growth. Stable microtubules are largely limited to main branches whereas dynamic microtubules are found in both main and higher order branches [9,14,15,37]. Nucleation of microtubules at branchpoints and anterograde

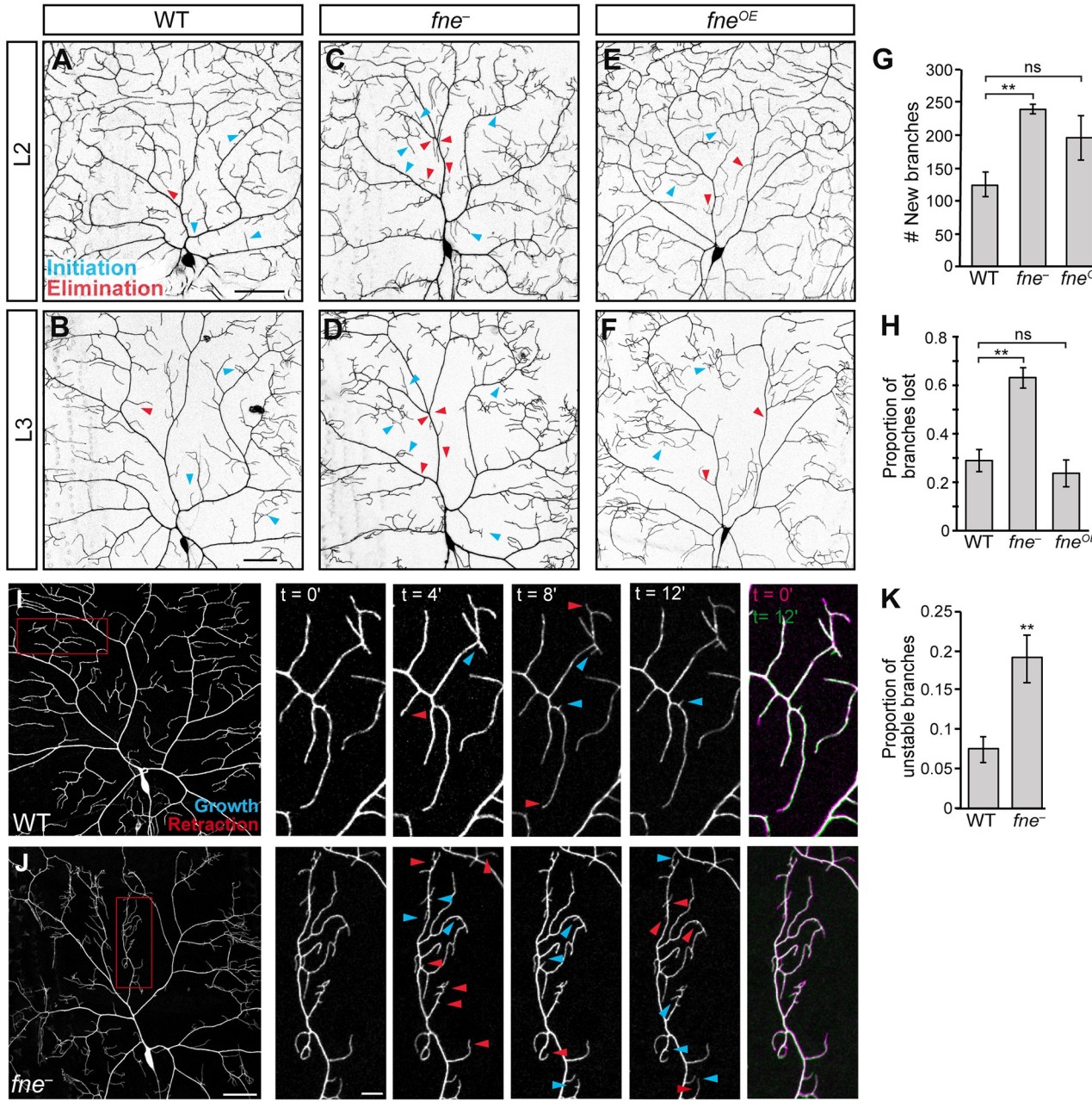

**Fig 2. Fne is required for branch stability.** (A-F) Confocal z-series projections of wild-type (WT; A, B), *fne⁻* (C, D) and *fne^OE* (E, F) neurons imaged at L2 (A-C) and then again at L3 (D-F). For *fne^OE*, *ppk-GAL4* was used to drive expression of *UAS-fne*. For all genotypes, *ppk-GAL4* was used to express *UAS-CD4-tdGFP* to mark the neurons. Arrowheads indicate new branches (blue) and lost branches (red) between L2 and L3. (G) Quantification of the number of new branches between L2 and L3 for WT (n = 6 neurons), *fne⁻* (n = 5 neurons), and *fne^OE* (n = 5 neurons). Values are mean ± s.e.m.; **p<0.01 compared to WT as determined by one-way ANOVA with Bonferroni-Holm *post hoc* test. (H) Quantification of the proportion of branches lost between L2 and L3 for neurons in (G). Values are mean ± s.e.m.; **p<0.01 compared to wild-type as determined by one-way ANOVA with Bonferroni-Holm *post hoc* test. (I, J) Time-lapse confocal z-series projections of WT (I) and *fne⁻* (J) class IV da neurons at late L3. Arrowheads indicate new/growing (blue) and lost/retracting (red) branches. (K) Proportion of branches undergoing initiation, elimination, growth or retraction events for WT (n = 4 neurons) and *fne⁻* (n = 3 neurons). Values are mean ± s.e.m.; **p<0.01 as determined by two-tailed Student's *t*-test; ns = not significant. Scale bars: 50 μm, 10 μm in enlargements.

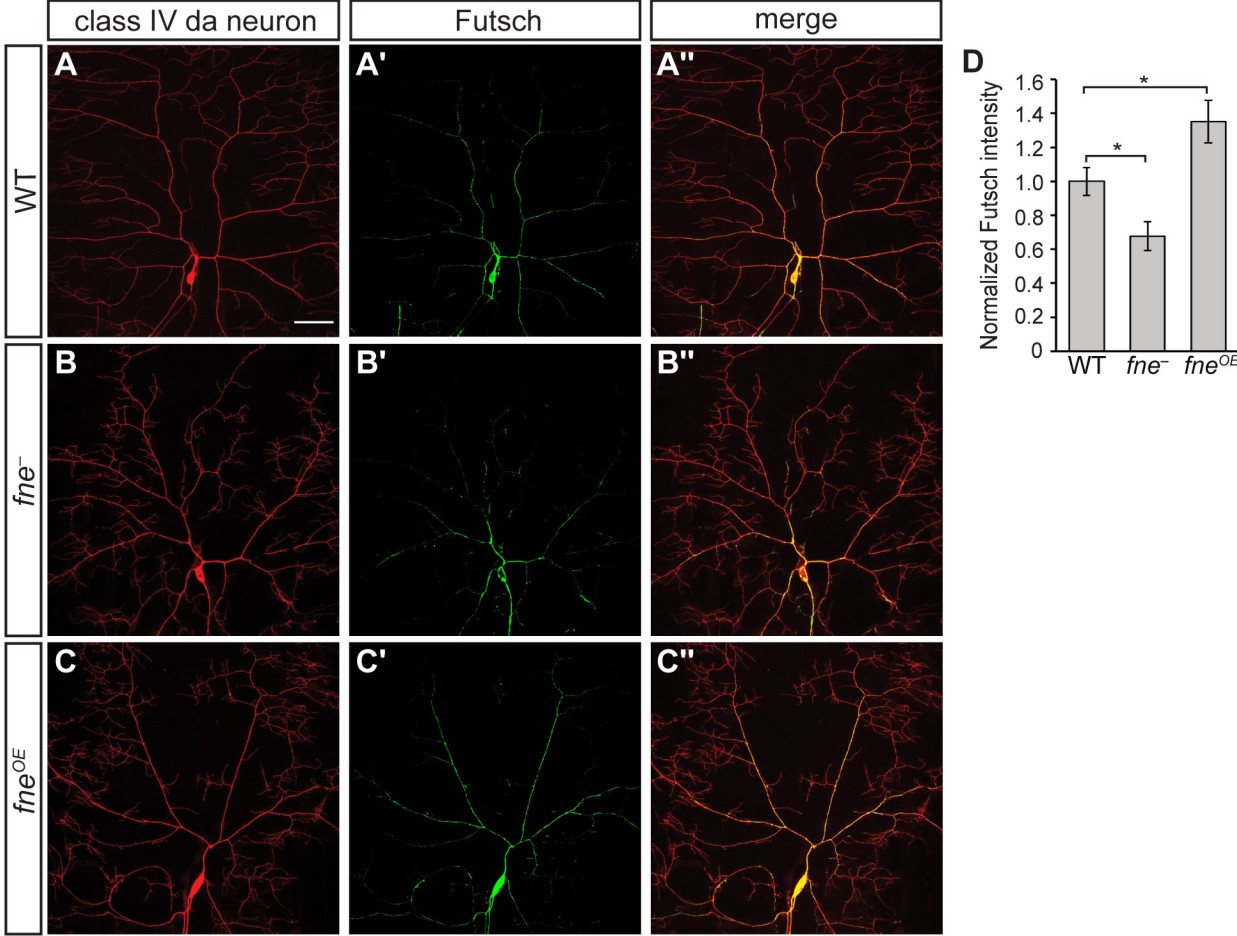

**Fig 3. Fne affects the microtubule content of class IV da neurons.** (A-A", B-B", C-C") Confocal z-series projections of wild-type (WT; A-A"), *fne⁻*(B-B") and *fne^OE* (C-C") neurons. *ppk-GAL4* was used to express *UAS-CD4-tdGFP* to mark the neurons and to drive expression of *UAS-fne*. Immunofluorescence was performed with anti-GFP to detect the neuronal marker (A-C) and anti-Futsch (A'-C') to label stable microtubules. The Futsch signal in other types of neurons has been masked to facilitate visualization of class IV da neurons. (D) Quantification of Futsch fluorescence intensity normalized to the intensity of the membrane marker for WT (n = 10 neurons), *fne⁻*(n = 11 neurons) and *fne^OE* (n = 11 neurons). Values are mean ± s.e.m.; *p<0.05 compared to WT, as determined by one-way ANOVA with Bonferroni-Holm *post hoc* test. Scale bar: 50 μm.

polymerization into new terminal branches promotes branch outgrowth and stabilization [15,37]. Notably, cytoskeletal composition, distribution, and dynamics differ among the four classes of da neurons and mutations in regulators that alter these properties can transform the morphology of one class of da neurons toward that of another [10,12,15,38]. Thus, we reasoned that the altered morphology of *fne* mutant neurons could result from altered cytoskeletal composition.

To determine whether dendritic microtubules are affected in *fne* mutant and overexpressing class IV da neurons, we analyzed the distribution of the *Drosophila* MAP1B homolog Futsch, which is a marker of stable microtubules [39,40]. Anti-Futsch immunofluorescence showed that the overall level of stable microtubules decreased by 32% in *fne⁻* arbors as compared to wild-type neurons (p<0.05) (Fig 3A–3A", 3B–3B" and 3D). Conversely, stable microtubule content increased by 35% throughout the proximal branches of *fne^OE* neurons (p<0.05) (Fig 3C–3C" and 3D). Similar trends were observed for acetylated tubulin, another marker of stable microtubules, although the differences did not reach significance (S4 Fig).

We also visualized actin content and distribution using LifeAct:GFP and moesin-GFP (*UAS-GMA*), but found no consistent differences between wild-type and either $fne^-$ or $fne^{OE}$ neurons. Thus, although we cannot rule out a role in actin organization or dynamics, our data point to a role for Fne in the regulation of stable microtubule content within the arbor.

## Role for Fne in integrin-mediated dendrite-ECM interaction

Early in development, the entire class IV da neuron arbor makes contact with the ECM, which provides both a stable support for branch extension and access to extracellular growth promoting cues [2,41,42]. As development proceeds, segments of the more proximal branches become enclosed by the surrounding epidermal cells, while the terminal branches remain in contact with the ECM [17,20]. Loss of dendrite-ECM interaction results in an increase in dendrite enclosure, defects in dendrite maintenance, and crossing of dendrites in 3-dimensional space [6,7,17]. The increased incidence of dendrite crossings in $fne^-$ neurons and dendrite maintenance defects (Figs 1 and 2) suggested a loss of ECM contact and increased enclosure. We therefore monitored dendrite enclosure, using an immunofluorescence assay that distinguishes enclosed from non-enclosed dendrites (see Materials and methods) [7,17]. In wild-type neurons, terminal branches remained largely in contact with the ECM. In contrast, we observed a 1.9-fold increase in the enclosure of terminal branches in $fne^-$ neurons (p<0.001) (Fig 4A–4C). These enclosed branches were often the site of dendrite crossing events (60% in WT, 80% in $fne^-$, p<0.01) (Fig 4D). Because the interaction of class IV da neuron dendrites with the ECM is mediated by integrins [6,7], we tested whether Fne and integrin interact genetically. Combinatorial overexpression of both the α-integrin subunit Multiple edematous wings (PS1; Mew) and the β-integrin subunit Myospheroid (βPS; Mys), which play a critical role in mediating dendrite-ECM interactions in class IV da neurons [6,7], rescued the crossing defects observed in $fne^-$ neurons (2.9 crossovers/1000 μm in $fne^-$, 0.8 crossovers/1000 μm in $fne^-$; *UAS-PS1, UAS-β PS*, p<0.01) (Fig 4E–4H). This result suggests that $fne^-$ neurons have weakened interactions with the ECM and increased contact with the overlaying epidermal cells. Decreased ECM interaction could contribute to the $fne^-$ terminal branch stabilization and crossing defects and thus the inability of these neurons to fill the requisite territory during larval growth.

Visualization of integrins in class IV da neurons is problematic because the proximity of class IV da neurons to the epidermis makes it difficult to distinguish between proteins mediating contacts between the epidermal cells and the ECM, and those that mediate dendrite-ECM contacts. We therefore tested the relationship between Fne and integrins using the *Drosophila* ovarian follicle cells, which constitute an easily visualized epithelial sheet that surrounds the oocyte and its sister nurse cells to create an egg chamber (S5 Fig). Ectopic expression of Fne in the follicle cells caused nurse cell death at later stages of egg chamber development (S5B and S5B' Fig). At earlier stages, however, anti-Mys immunofluorescence revealed a 2.5-fold increase in Mys levels and a less peripheral distribution than in wild-type follicle cells (p<0.001) (S5C, S5C', S5D, S5D' and S5E Fig), consistent with Fne acting to modulate cell-ECM adhesion. Additionally, ectopic expression of *fne* caused the typically cuboidal or columnar follicle cells to flatten, becoming more squamous in shape (S5C, S5C', S5D and S5D' Fig). The effects on cell shape appeared prior to nurse cell death, suggesting that these changes were not the result of apoptosis. Together, our results suggest that the mRNA targets of Fne modulate both the cytoskeletal composition and cell-ECM interactions to promote cell spreading, and in the case of class IV da neurons, stable, space-filling dendrite growth.

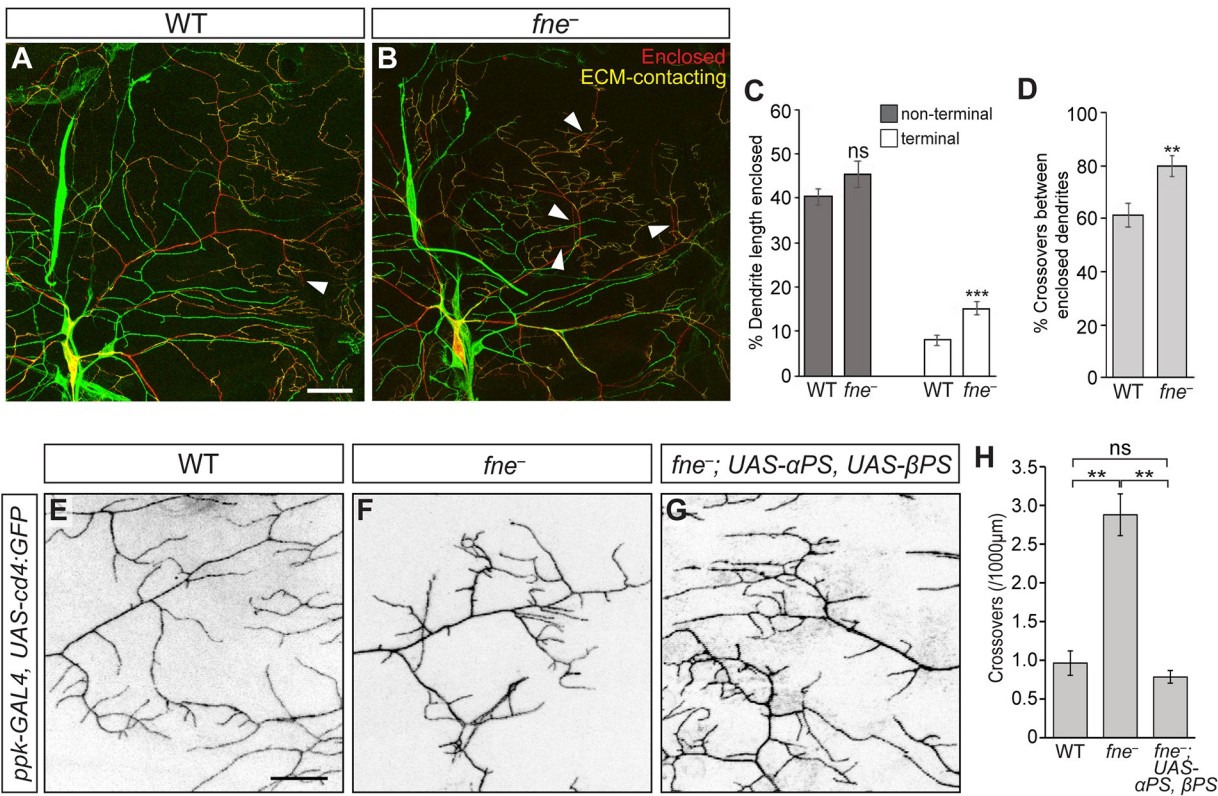

**Fig 4. Role for Fne in integrin-mediated dendrite-ECM interaction.** (A, B) Confocal z-series projections of the dorsal posterior quadrants of wild-type (WT; A) and *fne⁻* (B) L3 neurons. Class IV da neurons were marked with CD4-tdTom (red) and detected by anti-dsRed immunofluorescence. Non-enclosed dendrite segments were identified by anti-HRP immunostaining prior to permeabilization (green; see Materials and methods). In the merged images, non-enclosed, ECM-contacting dendrite segments appear yellow whereas enclosed segments appear red. Other neurons in the field detected by anti-HRP appear green only. White arrowheads denote dendrite crossing events. (C) Quantification of enclosure in non-terminal dendrites and terminal dendrites in WT and *fne⁻* neurons (n = 10 neurons each). (D) Quantification of the proportion of all crossovers that occur in regions of enclosure (n = 10 neurons each). (E-G) Confocal z-series projections of (E) WT, (F) *fne⁻* and (G) *fne⁻*; *UAS-αPS1*, *UAS-β PS* neurons. *ppk-GAL4* was used to drive expression of *UAS-CD4-tdTom* to mark the neurons and *UAS-αPS1*, *UAS-β PS*. (H) Quantification of dendrite crossing events (n = 8 neurons each). Values are mean ± s.e.m.; **p<0.01, ***p<0.001 as determined by two-tailed Student's *t*-test (C, D) or ANOVA with Bonferroni-Holm *post hoc* test (H). Scale bars: (A) 50 μm, (D) 20 μm.

### TRIBE identifies transcripts encoding cytoskeletal regulators and cell adhesion proteins as Fne targets

To identify mRNAs directly bound by Fne, we took advantage of the TRIBE method [43], which marks *in vivo* targets of RNA-binding proteins by novel RNA editing events. Fne was fused to the catalytic domain of the RNA editing enzyme ADAR (ADAR$_{CD}$) (S6A Fig), which deaminates adenosine to produce inosine. Inosine is read as guanosine by reverse-transcriptase, resulting in A-to-G transitions that can be identified by RNA sequencing (RNA-seq). Consequently, RNAs bound by Fne-ADAR$_{CD}$ are revealed by such editing events. Because class IV da neurons are relatively few in number and difficult to isolate, we performed TRIBE in *Drosophila* S2 cells. Although Fne is not endogenously expressed in S2 cells, we assumed, based on the above findings, that many Fne targets would not be neuron-specific. cDNA libraries were prepared using RNA isolated from stable cell lines expressing either Fne-ADAR$_{CD}$ or ADAR$_{CD}$ alone under the control of a metallothionine promoter (S6B Fig). RNA-seq analysis identified 147 putative Fne targets that contained one or more A-to-G edits that were present at the exact same nucleotide in two biological replicates (Fig 5A, S1 and

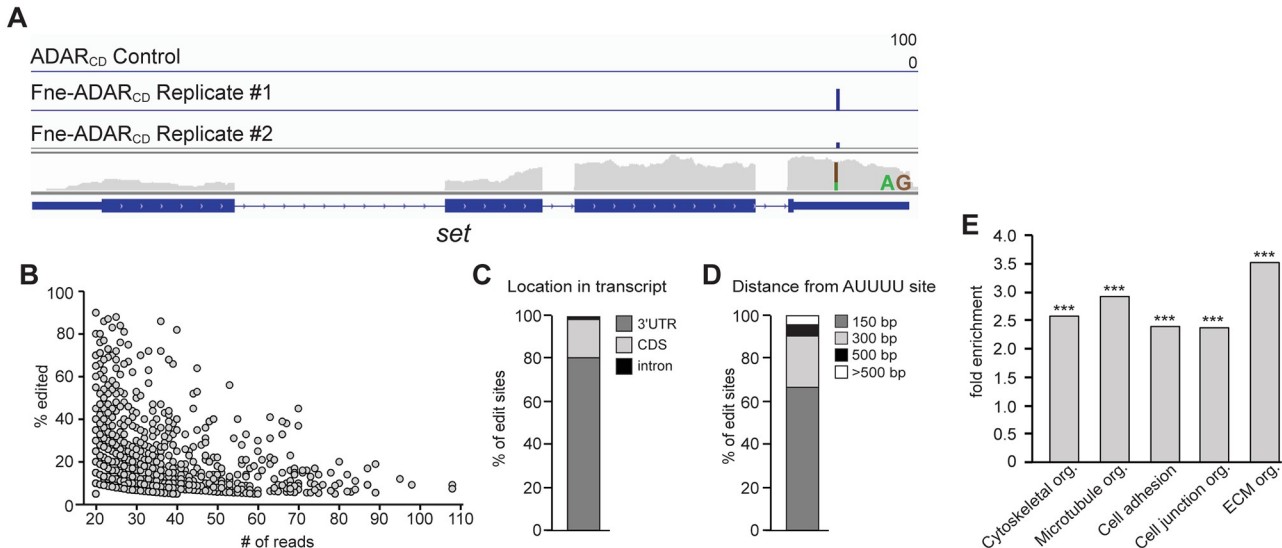

**Fig 5. TRIBE analysis in *Drosophila* S2 cells identifies Fne target transcripts.** (A) A sample gene, *set*, showing edits detected in control cells expressing ADAR$_{CD}$ and in two biological replicates from cells expressing Fne-ADAR$_{CD}$. mRNA expression level is shown below the edit tracks. Editing events are indicated by the blue bars, with the height of the bar representing the editing percentage for a given site. The editing percentage is a measure of the number of times a site was edited relative to its total read count. For editing events, thresholds of 5% editing and 20 reads were applied. (B) Editing percentage shown as a function of the number of reads at a given site, indicating no relationship between editing frequency and transcript abundance. (C) Quantification of the frequency of edited sites in the indicated transcript locations. (D) Quantification of the proximity of edited sites to an AUUUU motif. (E) Quantification of the overrepresentation of certain GO terms in targets obtained for Fne. ***p<0.001 as determined by Fisher's exact test.

S2 Tables). We found highly significant overlap between the two replicates (p = 7.5 x 10$^{-257}$) and there was no correlation between the editing frequency of a nucleotide and the number of times that nucleotide was read (Fig 5B), suggesting that editing frequency did not simply reflect mRNA abundance. Fne was previously shown to bind the *elav* 3'UTR in a region containing four AU$_{4-6}$ motifs [44]. Notably, 80% of the Fne-dependent edits were located in 3'UTRs and 89% were within 300 nt of an AUUUU motif (Fig 5C and 5D). Additionally, we verified a subset of TRIBE candidates by RNA co-immunoprecipitation with Fne-V5 (S7 Fig), suggesting that the TRIBE assay detected *bona fide* Fne targets. GO term analysis of all potential Fne targets identified showed an enrichment for RNAs encoding proteins involved in cytoskeletal organization, cell migration and adhesion (Fig 5E). These results are consistent with our experimental evidence that Fne acts to modulate both cytoskeletal composition and dendrite-ECM adhesion to promote stable dendrite growth. Fne also bound transcripts encoding proteins involved in RNA-binding and post-transcriptional regulation (S2 Table and S7 Fig), suggesting Fne could have additional indirect effects. We note that only a subset of Fne targets can be identified by TRIBE due to this method's limited sensitivity. However, targets identified for other RBPs were shown to correspond to high-confidence CLIP targets [43], suggesting that transcripts identified by TRIBE represent stably bound, biologically-relevant targets.

## Neuronal and epidermal Basigin is critical for proper arbor morphology

We chose to further validate one of the TRIBE targets verified by RNA co-immunoprecipitation, *basigin* (*bsg*), which had not previously been implicated in class IV da neuron development (Fig 6A). *bsg* encodes an immunoglobulin superfamily transmembrane glycoprotein that has been shown to promote cytoskeletal reorganization and interact with integrin to promote cell shape changes in cultured cells [45]. Since the TRIBE analysis was performed in S2 cells,

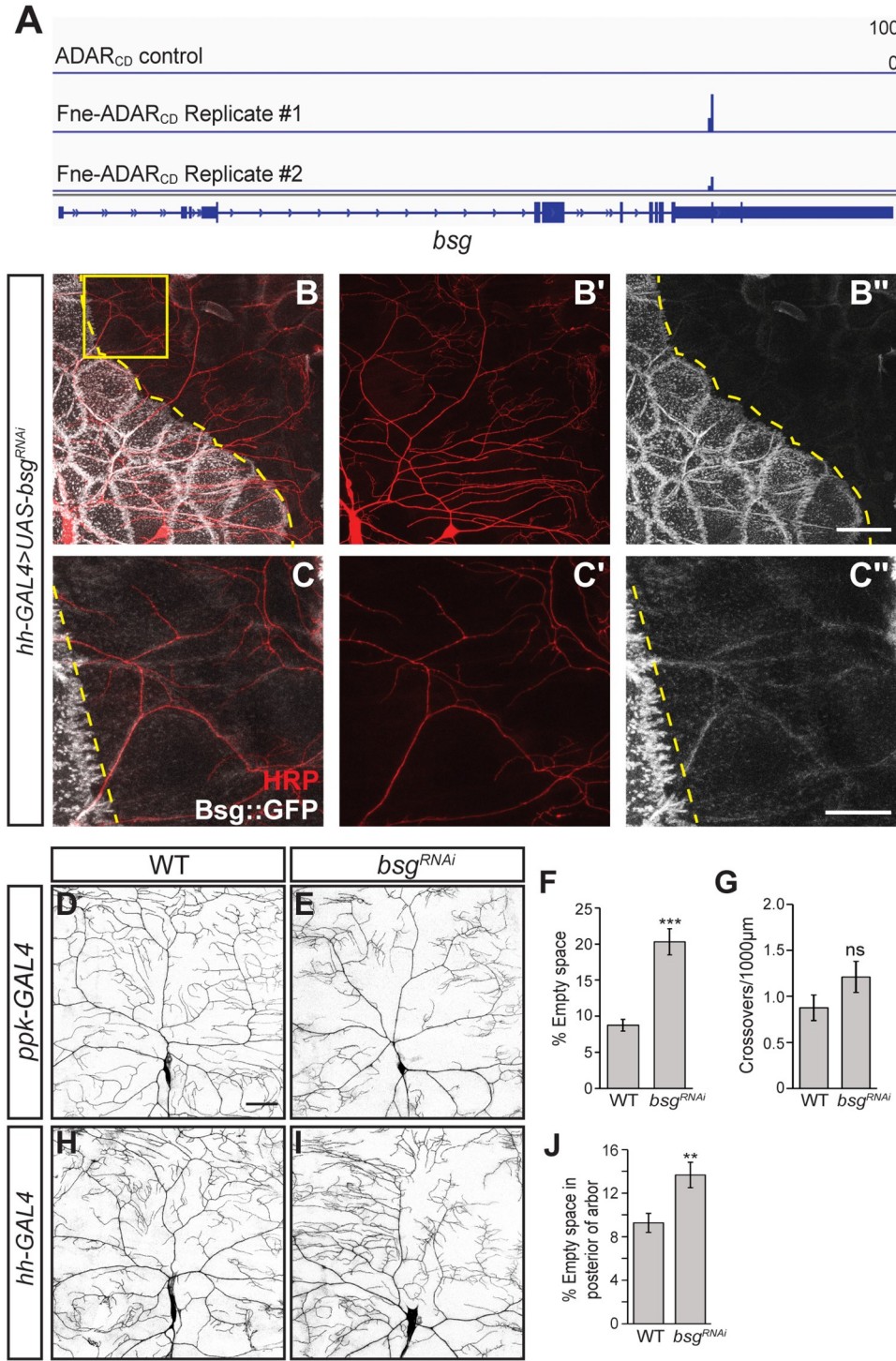

**Fig 6. *bsg* is required in both the neuron and the epidermis for class IV da neuron development.** (A) *bsg* transcript with edits detected in control cells expressing ADAR$_{CD}$ and in two biological replicates from cells expressing Fne-ADAR$_{CD}$. Editing events are indicated by the blue bars, with the height of the bar representing the editing percentage for a given site. (B-B") Confocal z-series projection of a neuron from a larva expressing *UAS-bsgRNAi* (VDRC 105293) in the posterior of the epidermis (to the right of the dotted yellow line) using *hh-GAL4*. Anti-HRP was used to visualize neurons and Bsg::GFP was detected by anti-GFP immunofluorescence. (C-C") show enlargements of the boxed area in (B-B"). (D, E) Confocal z-series projections of neurons from wild-type (WT; D) and *bsg^{RNAi}* (E) larvae. *ppk-GAL4* was used to express *UAS-CD4-tdTom* to mark the neurons and to drive expression of *UAS-bsgRNAi* (VDRC 105293).

(F) Quantification of empty space in the arbor for WT (n = 10 neurons) and $bsg^{RNAi}$ (n = 11 neurons). (G) Quantification of crossing events for neurons in (F). (H, I) Confocal z-series projections of neurons from a WT larva (H) and a larva expressing *UAS-bsgRNAi* in the posterior portion of the epidermis driven by *hh-GAL4* (I). Neurons were marked using *ppk-CD4-tdGFP*. (J) Quantification of empty space in the posterior third of the arbor for WT (n = 22 neurons) and $bsg^{RNAi}$ (n = 17 neurons) in (H, I). Values are mean ± s.e.m.; \*\*p<0.01, \*\*\*p<0.001 as determined by two-tailed Student's *t*-test. Scale bars: (B", D) 50 μm (C") 20 μm.

we first asked whether Bsg is expressed in class IV da neurons and if it is required for class IV da neuron development. To analyze Bsg expression, we used a GFP protein trap insertion in the *bsg* gene (Bsg::GFP) [46]. This protein trap is expressed from the *bsg* promoter, providing a readout of endogenous Bsg expression. We found that Bsg is enriched at epidermal cell borders and in long tracks along the primary and secondary branches of class IV da neurons (Figs 6B–6B" and Fig 7A–7A'''). These tracks could represent Bsg in the overlying epidermal cells, in the neuron, or in both. To distinguish among these possibilities, we first assayed the effect of knocking down *bsg* in epidermal cells on the appearance of Bsg tracks. *UAS-bsgRNAi* was expressed in epidermal cells located in the posterior of each larval segment (i.e., over the posterior portion of the dendritic arbor) using the *hh-GAL4* driver [18]. This eliminated most of the epidermal Bsg::GFP fluorescence. Dendrite tracks persisted, albeit at a lower intensity than the tracks found in the anterior portion of the arbor (Fig 6B–6B" and 6C–6C"), suggesting that the Bsg tracks visible along the dendrites are created by both neuronal and epidermal Bsg. To further confirm this and to determine whether Bsg is required in one or both cell types, we analyzed the effects of either neuronal or epidermal-specific *bsg* RNAi on dendrite morphology. Similarly to *fne* mutant and overexpressing neurons, knockdown of *bsg* in class IV da neurons using *ppk-GAL4* led to loss of dendritic field coverage (empty space = 9% in WT, 20% in $bsg^{RNAi}$, p<0.001) (Fig 6D–6F). Epidermal knockdown of *bsg* using *hh-GAL4* led to similar space-filling defects in the posterior portion of the arbor, which underlies the epidermal expression domain of *hh-GAL4* (empty space = 9% in WT, 14% in $bsg^{RNAi}$, p<0.01) (Fig 6H–6J) [18]. A second independent *UAS-bsgRNAi* line produced similar results (S8 Fig) and both were used in subsequent experiments. Together, these results indicate that Bsg tracks comprise dendrite and epidermal Bsg, and that Bsg is required in both cell types for proper dendrite morphogenesis.

## Fne regulation of *bsg* promotes the space-filling morphology of class IV da neurons

To determine whether *bsg* is a target of Fne in class IV da neurons, we analyzed Bsg expression in *fne* mutant neurons using the Bsg::GFP protein trap. In *fne*⁻ neurons, Bsg tracks along dendrites were eliminated (Fig 7A–7A''' and 7B–7B'''). While these tracks consist of epidermal and neuronal Basigin (Fig 6), the fact that Fne is neuron-specific and can bind *bsg* directly suggests that Fne positively regulates Bsg levels in the neuron. This result also indicates that eliminating neuronal Bsg leads to a failure to recruit epidermal Bsg, suggesting that there are homophilic Bsg interactions between dendrites and epidermal cells. Furthermore, positive regulation by Fne predicts that reducing *bsg* levels should suppress defects caused by *fne* overexpression. Indeed, *bsg* RNAi partially suppressed the loss of coverage associated with $fne^{OE}$ (empty space = 22% in $fne^{OE}$, 18% in $fne^{OE}/bsg^{RNAi}$, p<0.05) (Fig 7C–7G) but had no effect on *fne*⁻ neurons (S8 Fig), suggesting that defects observed in $fne^{OE}$ neurons can be attributed in part to altered levels of Bsg.

We next investigated whether the effect on microtubule content and the integrin-mediated ECM adhesion defects observed when *fne* function is altered could result from misregulation of Bsg. Similarly to *fne*⁻ neurons, neuron-specific *bsg* RNAi caused a 33% decrease in Futsch

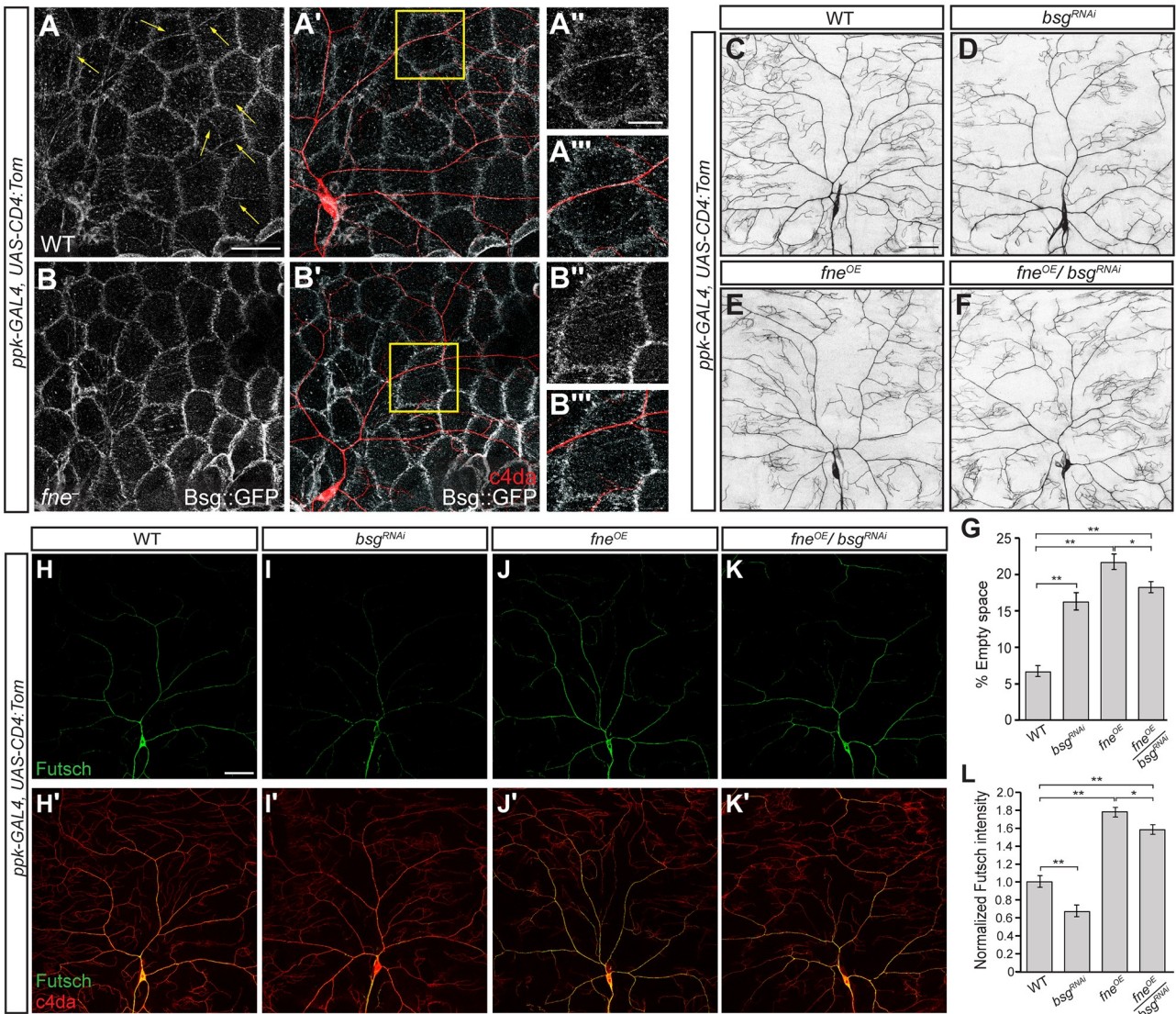

**Fig 7. Regulation of *bsg* by Fne promotes space-filling dendrite growth through cytoskeletal organization.** (A, A', B, B') Confocal z-series projections of wild-type (WT; A, A') and *fne⁻* (B, B') neurons. *ppk-GAL4* was used to express *UAS-CD4-tdTom* to mark the neurons (red in merged images A' and B'). Bsg::GFP was detected by anti-GFP immunofluorescence. Yellow arrows in (A) mark tracks of Bsg::GFP along the dendrites. Enlargements of the boxed areas in A' and B' are shown in A", A''' and B", B'''. (C-F) Confocal z-series projections of WT (C) *bsg^RNAi* (D), *fne^OE* (E) and *fne^OE*/*bsg^RNAi* (F) neurons. *ppk-GAL4* was used to drive expression of the *UAS-fne* and *UAS-bsgRNAi* (BDSC 52110) transgenes. For all genotypes, *ppk-GAL4* was used to express *UAS-CD4-tdTom* to mark the neurons. (G) Quantification of empty space for WT (n = 12 neurons), *bsg^RNAi* (n = 19 neurons), *fne^OE* (n = 15 neurons), and *fne^OE*/*bsg^RNAi* (n = 20 neurons). (H-K) Anti-Futsch immunofluorescence to detect stable microtubules for the genotypes indicated in (C-F). The Futsch signal in other types of neurons has been masked to facilitate visualization of class IV da neurons. Merged images of Futsch and CD4-tdTom for the indicated genotypes are shown in H'-K'. (L) Quantification of Futsch intensity normalized to the neuronal membrane marker for WT (n = 15 neurons), *bsg^RNAi* (n = 10 neurons), *fne^OE* (n = 11 neurons), and *fne^OE*/*bsg^RNAi* (n = 12 neurons). Values are mean ± s.e.m.; *p<0.05, **p<0.01 as determined by one-way ANOVA with Bonferroni-Holm *post hoc* test. Scale bars: 50 μm.

levels throughout the arbor (p<0.01). Furthermore, neuronal knockdown of *bsg* modestly suppressed the elevated dendritic Futsch levels caused by *fne^OE* (11% decrease, p<0.05) (Fig 7H–7K, 7H'–7K' and 7L). Bsg tracts do coincide with regions of dendrite enclosure (S9 Fig). However, knockdown of *bsg* in class IV da neurons did not affect isoneuronal dendrite crossing events (Fig 6G). Thus, Bsg likely does not play a role in regulating integrin-mediated dendrite adhesion to the ECM and the dendrite crossing events observed in *fne⁻* neurons must result

from misregulation of other Fne targets. Together, these results suggest that Bsg acts as a downstream effector of Fne in class IV da neurons to promote the cytoskeletal organization required for space-filling dendrite growth, and that the coverage defects observed in *fne* mutant neurons are due in part to decreased Bsg expression.

## Discussion

Dendrite branching is a dynamic process that depends heavily on the regulation of cytoskeletal organization and dendrite-substrate interactions. Class IV da neurons depend on Fne throughout development as the rapidly growing epidermis demands both the extension of existing branches and elaboration of new branches in order to maintain coverage. Our results indicate that Fne regulates targets involved in cytoskeletal organization and dendrite-ECM interactions, leading to stable branch growth and arbor elaboration. Whereas terminal branches in wild-type neurons are largely stable by the end of larval development, these branches remain highly dynamic in *fne⁻* neurons likely due to weakened interactions with the ECM, although we cannot rule out additional effects on actin dynamics. These weakened dendrite-ECM interactions as well as the decrease in stable microtubules in *fne⁻* neurons prevent long-term branch stabilization and, consequently, field coverage is not maintained as the larva grows.

The ability of both loss and overexpression of *fne* to cause space-filling morphology defects despite their opposing effects on microtubule content is consistent with previous work showing that increased and decreased microtubule stability in class IV da neurons can cause similar effects on branching [15]. Loss or overexpression of *fne* also has consequences during pupariation, when class IV da neuron dendrites are pruned back to the soma. As microtubule breakdown is an important first step in pruning [47,48] and is linked to dendrite thinning and destabilization of the dendritic membrane [48], the pruning defects in *fne* mutant and overexpressing neurons may arise from the same effects of Fne on microtubule composition observed in larval neurons. Destabilization of microtubules in *fne⁻* neurons could lead to a premature initiation of the pruning process. Conversely, increased microtubule stability in *fne^{OE}* neurons could lead to a delay in this process.

The identification of RNAs encoding cytoskeletal regulators and cell adhesion molecules like Bsg as targets of Fne provides insight into to how Fne may coordinate inputs to dendrite patterning. The requirement for Bsg in both the neuron and epidermis for proper class IV da neuron morphogenesis supports a role for Bsg as an effector of Fne in mediating dendrite communication with the overlying epidermal cells. Recent work has implicated epidermally-derived signals in class IV da neuron space-filling morphology [18,19]. Among these, Syndecan, a heparin sulfate proteoglycan (HSPG) on the surface of epidermal cells, promotes microtubule stabilization in higher order dynamic branches in order to promote the space-filling morphology of class IV da neurons [18], although how Syndecan communicates with the neuron is currently unknown. In T cells, the mammalian homolog of Bsg, cluster of differentiation 147 (CD147), forms a complex with Syndecan-1 in *cis* [49]. Similarly, epidermal Bsg could interact with epidermal Syndecan and bind to neuronal Bsg to signal from the epidermis to the neuron to promote branch stabilization. Our finding that Bsg is required for space-filling growth but not for regulating terminal branch dynamics fits well with results from previous work on Syndecan showing that the microtubule stabilization promoted by HSPG is required for long-term branch stabilization but not short-term branching dynamics [18] and suggests that the two molecules may interact. Furthermore, our finding that knockdown of *bsg* reduces stable microtubule content similarly to mutation of *fne* and can partially ameliorate the increase in stable microtubules caused by *fne* overexpression supports the idea that Fne

activation of Bsg expression during larval growth facilitates dendrite-epidermal communication that in turn impacts the dendritic microtubule cytoskeleton. This role of Fne in epidermal-neuronal control of space-filling dendrite growth was not revealed by our initial phenotypic analysis, but only through TRIBE identification of Fne target RNAs.

RNA-binding proteins typically have numerous targets and the TRIBE data indicate that this is the case for Fne. Thus, it is not surprising that regulation of *bsg* accounts for only a subset of defects observed in *fne⁻* neurons. Evidence that Bsg colocalizes with integrins in cultured *Drosophila* cells and in the *Drosophila* retina [45], and coimmunoprecipitates with integrin from mammalian cells [50] initially suggested that Bsg might function in the integrin-dependent adhesion of class IV da neurons to the ECM. Our results, however, do not support such a role, as dendrite crossings–which are a consequence of loss of dendrite-ECM adhesion–were not affected by neuronal *bsg* RNAi. Although the high false-negative rate of TRIBE may have precluded the identification of *mys* as a target of Fne, our analysis did identify several candidates that could be effectors of dendrite-ECM adhesion. One potential candidate is *sema-1b*, since another semaphorin, *sema-2a*, has been shown to promote integrin-mediated dendrite-ECM adhesion in class IV da neurons [51]. Another candidate, 14-3-3ζ, has been shown to directly interact with L1 cell adhesion molecule (L1CAM) to limit neurite outgrowth in mice [52]. In class IV da neurons, the L1CAM homolog Neuroglian is a component of the enclosure complex [53], suggesting that excess 14-3-3ζ in *fne⁻* neurons might lead to increased enclosure and loss of contact with the ECM. Further analysis of these targets, and their interactions with Fne, could elucidate the ways in which Fne mediates dendrite growth along the ECM.

Although previous work has uncovered many transcriptional regulators of class IV da neuron development, less is known about the post-transcriptional mechanisms that govern the morphology of these neurons. In contrast to transcriptional regulation, post-transcriptional regulation allows for rapid and localized control. Such features are particularly important in neurons, which must respond rapidly to a variety of extracellular and developmental cues and whose dendrites can extend long distances from the soma. By regulating numerous, functionally-related transcripts, a single RBP can efficiently promote synchronized control over multiple inputs that impact neuronal patterning. In this manner, Fne may ensure that both the cytoskeletal organization and dendrite-substrate interactions required for stable, space-filling dendrite growth are regulated in tandem. How Fne acts on its targets, however, is poorly understood. Although Elav/Hu proteins have been shown to function at almost every stage of RNA metabolism, the somatic, cytoplasmic localization of Fne in class IV da neurons suggests that it functions in regulating transcript stabilization or translation. Furthermore, as Fne contains only RNA recognition motifs and no other known functional domains, it likely acts by recruiting other proteins to its target transcripts. Identification of the protein interacting partners of Fne will shed light on the molecular mechanism(s) by which Fne controls its various targets.

The space-filling defects observed in *fne⁻* class IV da neurons are similar to the neuronal defects observed in *HuD* knockout mice. Loss of HuD led to decreased total branch length and reduced arbor complexity in neurons in the lower layer of the neocortex and the CA3 region of the hippocampus, indicating a role for HuD in the expansion of these neurons early in development [31]. These results suggest a common role for the two homologous proteins. The defects we observe when *fne* is overexpressed may shed light on the phenotype of metastatic cancers including small cell lung carcinoma (SCLC) and neuroblastoma that express the neuron-specific HuD protein [25]. SCLC cells take on properties of migrating neuroblasts, extending microtubule-rich axon-like projections that increase their ability to metastasize [54]. Furthermore, integrin-mediated ECM interactions have been previously shown to increase

metastasis and migration in SCLC cells [55] and play an important role in neuroblast migration [56]. In class IV da neurons, *fne* overexpression prevented branching along the main dendrites and forced branching to occur at the periphery of the arbor, likely through increased microtubule stabilization and altered dendrite-ECM contacts. Ectopic expression of *fne* in epithelial cells led to alterations in integrin expression and distribution. Additionally, expression of *fne* caused these normally cuboidal epithelial cells to become squamous, suggesting that the changes in cytoskeletal composition and cell-ECM interactions coordinated by Fne promote cell spreading. The ability of Fne to regulate both cytoskeletal organization and integrin-mediated cell-ECM interactions suggests that HuD may drive similar processes that produce the neuron-like morphology and migratory properties of SCLC cells. Thus, whereas the cellular behaviors promoted by Fne support the unique space-filling morphology of class IV da neurons, they could result in untoward effects in epithelial cells when expression of Fne homologs is dysregulated.

## Materials and methods

### Fly stocks

The following mutant and transgenic lines were used: $fne^{z40}$ [26], *UAS-fne* (II) [26], *UAS-fne* (III) [26], *UAS-fneRNAi* (VDRC 101508), *UAS-fneRNAi* (TRiP; BDSC 53340), *Bsg::GFP* [46], *tj-GAL4* (*Drosophila* Resource Center, Kyoto), $GAL4^{69B}$ (BDSC 1774), *UAS-bsgRNAi* (VDRC 105293), *UAS-bsgRNAi* (TRiP; BDSC 52110), and $GAL4^{58A}$, *ppk-CD4-tdTom* [20]. Combinatorial overexpression of integrin subunits was performed using a *UAS-PS1*, *UAS-βPS* recombinant chromosome provided by K. Broadie (Vanderbilt University). Class IV da neurons were visualized using *ppk-GAL4, UAS-CD4-tdGFP* [57], *ppk-GAL4, UAS-CD4-tdTom* [57], *ppk-CD4-tdGFP* [58] or *ppk-CD4-tdTom* [58]. Crosses involving *UAS-RNAi* lines, as well as their controls, were performed at 29°C to increase *GAL4/UAS* efficiency. All other crosses were performed at 25°C.

### Immunostaining of neurons

Immunostaining was performed on larval fillets as previously described [57]. The CD4-GFP membrane marker and Bsg::GFP were detected using 1:500 Alexa Fluor 488 anti-GFP antibody (Life Technologies, A-21311). The CD4-Tom membrane marker was detected using 1:500 rabbit anti-RFP antibody (Rockland ImmunoChemicals, 600-401-379). Fne expression was visualized using 1:100 anti-Fne [28]. Stable microtubules were detected using 1:20 anti-Futsch monoclonal antibody 22C10 (Developmental Studies Hybridoma Bank [DSHB]) or 1:500 anti-acetylated tubulin antibody (Sigma T6793). For visualization of microtubules or Bsg::GFP, the larval body wall muscles were removed as described previously [59]. The following secondary antibodies (Life Technologies) were used at 1:500: AlexaFluor 488 goat anti-rat (A-11006), AlexaFluor 568 goat anti-rabbit (A11011), AlexaFluor 488 goat anti-mouse (A-11001) and AlexaFluor 568 goat anti-mouse (A-21235). All antibody incubations were done in 5% normal goat serum (NGS) in PBS/0.3% TritonX-100 for either 1.5 h at RT or overnight at 4°C.

Confocal imaging was performed on either a Leica SPE laser scanning confocal microscope (20x/0.7 numerical aperture (NA) air objective or 40x/1.25 NA oil objective) or a Leica SP5 laser scanning confocal microscope (40x/1.25 NA oil objective). ddaC neurons in segments A3-A5 were imaged, with the soma placed in a similar location for all images in an experiment. The surfaces function in Imaris (Oxford Instruments) was used to show only ddaC-colocalized Futsch signal, as indicated in the legends for Figs 3 and 7.

## Analysis of dendrite morphology

The number of dendritic terminal branches was counted manually in Fiji. To measure total branch length and terminal branch length, neurons were traced using the NeuronJ plugin in Fiji. To measure the field coverage, a grid of 18 μm x 18 μm unit squares was overlaid on the image of the arbor and the number of empty squares was counted. The distribution of branches within the arbor was determined using the Fiji Sholl plug-in on traced images to measure the number of branches at a given distance from the soma [14]. For dendrite crossing quantification, the number of crossovers was normalized to the total branch length.

## Fluorescence intensity quantification

Fluorescence intensity was measured in Fiji using maximum intensity projection images. The average intensity of a line drawn through the dendrite of interest and a line drawn in the adjacent background region was determined for both Futsch or acetylated tubulin and the neuronal marker. The background intensity was subtracted and the ratio between Futsch or acetylated tubulin intensity and the intensity of the neuronal marker was determined.

## Analysis of pupal neurons

To analyze dendrite morphology at 16 h APF, pupae were prepared as previously described [60]. For 7 h APF samples, the protocol was modified slightly due to the fragility of the tissue. The anterior and posterior ends of each pupa were cut off and the pupa was cut along the coronal plane to remove the ventral portion. The pupa was then pinned and the internal tissue was carefully removed using forceps and gentle washing with a pipette. The tissue was then fixed, followed by anti-GFP immunofluorescence and imaging as described above. The number of primary and secondary dendrites attached to the soma was manually counted in ImageJ.

## Live imaging and analysis of branching dynamics

For live imaging, L2 larvae were mounted in 90% glycerol and a single neuron was imaged using a Leica SPE confocal microscope (20x/0.7 NA air objective). The larva was then placed back in the food and allowed to develop for 48 hours. The larva was mounted again in 90% glycerol and the same neuron was re-imaged. For each neuron, the number of branches completely eliminated and the number of new branches formed between L2 and L3 were quantified. For branch elimination, an area of the arbor was chosen in the L2 sample. The number of branches from that area that were lost by L3 was counted and divided by the total number of branches in that area at L2 to obtain the percentage of branches lost. Branch formation was quantified throughout the arbor as branches that were present at L3 but not at L2. For time-lapse analysis of branch dynamics, the larva was mounted as described above and a z-series of a single neuron was captured every 4 min for a total of 12 min (Leica SPE 20x/0.7 NA air objective). Initiation, elongation, elimination and retraction events were counted.

## Visualization and quantification of dendrite enclosure

Class IV da neurons were labeled using *ppk-GAL4* to express *UAS-CD4-tdTom*. Immunostaining to detect dendrite enclosure was performed as previously described [17,59]. Briefly, after dissection and muscle removal, larval fillets were fixed and incubated with 1:200 Alexa Fluor 488 goat anti-horseradish peroxidase (HRP) antibody (Jackson ImmunoResearch #123-545-021) without permeabilization. Under these conditions, non-enclosed regions of dendrites are more accessible than enclosed regions to anti-HRP, which binds neuronal cell surface antigens, and are thus preferentially labeled. After removal of the anti-HRP antibody, the tissue was

permeabilized and incubated with 1:500 rabbit anti-dsRed (Clontech, #632496) followed by 1:500 Alexa Fluor 568 goat anti-rabbit to label the entire class IV da neuron arbor. Dendrites were manually traced using the NeuronJ plugin in Fiji to quantify total branch length (anti-dsRed channel) and the total length of enclosed segments (low signal in anti-HRP channel). Analysis was restricted to the dorsal-posterior quadrant of ddaC neurons and dendrites innervating segment boundaries were omitted from the analysis.

## Immunostaining of follicle cells

Newly hatched females were kept at 25˚ C for 4 days, and were then fed on yeast paste for one additional day at 25˚ C. Ovaries were dissected and fixed as previously described [61]. After rehydration into PBS/0.1% Tween (PBST), ovaries were blocked in Image-iT FX (Thermo Fisher Scientific) for 30 min at RT, and then blocked in 5% NGS/ PBST for 1 h at RT. Mys was detected using 1:25 anti-Mys monoclonal antibody CF.6G11 (DSHB) and 1:500 AlexaFluor 568 goat anti-mouse, with antibody incubations performed in 5% NGS/ PBST for either 1.5 h at RT or overnight at 4˚ C. DAPI (Molecular Probes) was used to visualize DNA. Ovaries were mounted in Vectashield and imaged on a Leica SP5 laser scanning confocal microscope (40x/ 1.25 NA oil objective).

Mys intensity was measured in Fiji using single confocal sections. For each egg chamber, anti-Mys fluorescence intensity in individual follicle cells was measured using the integrated density function. A similar background intensity measurement was made for an area adjacent to each cell and subtracted from the cell measurement to obtain a Mys intensity value. Three to four cells were quantified from four to five egg chambers for each genotype.

## Plasmid construction

KpnI and NotI restriction sites were added to the 5' and 3' ends of the *fne* coding sequence (gift of M. Soller, University of Birmingham UK), respectively, using PCR. pMT-Hrp48-A-DAR$_{CD}$-V5 (Addgene) was digested using the same enzymes and the *fne* coding sequence was ligated into the vector backbone. To make the ADAR$_{CD}$ only control, all but the first 36 nucleotides of the *fne* coding sequence was removed from the pMT-Fne-ADAR$_{CD}$-V5 plasmid by digestion with BamHI and NotI. After end-filling with Klenow, the plasmid was recircularized with T4 DNA ligase. To generate the pMT-Fne-V5 construct used for the RNA-IP, pMT-Fne-ADAR$_{CD}$-V5 was cut with PshAI and ApaI. After end-filling with Klenow, the plasmid was recircularized with T4 DNA ligase.

## Generation of cell lines and immunoblotting

*Drosophila* S2 cells were maintained at 25˚ C in Schneider's medium (Life Technologies) supplemented with 10% fetal bovine serum (Life Technologies), 1% GlutaMAX (Life Technologies) and 1% penicillin/streptomycin (Life Technologies). The pMT-Fne-ADAR$_{CD}$-V5, pMT-ADAR$_{CD}$-V5, and pMT-Fne-V5 plasmids were transfected into the cells using Effectene Transfection Reagent (Qiagen) and stable cell lines were obtained by selecting for blasticidin resistance. Protein expression was induced by addition of CuSO$_4$ at a final concentration of 0.5 mM. After 24 h, the cells were harvested and lysed in RIPA buffer (150 mM NaCl, 5 mM EDTA, 50 mM Tris pH 8.0, 1% NP-40, 0.5% deoxycholate, 0.1% SDS). Proteins were detected by western blotting as described [57] with 1:1000 anti-V5 antibody (Thermo Fisher Scientific, MA5-15253). Kinesin heavy chain (Khc) was used as a loading control and detected with anti-Khc antibody (1:10,00; Cytoskeleton).

## TRIBE assay

Expression of Fne-ADAR$_{CD}$ and ADAR$_{CD}$ was induced for 24 h. Cells were harvested and then resuspended in lysis buffer (10 mM Tris HCl pH 7.5, 150 mM NaCl, 0.5 mM EDTA, 0.5% NP-40). Samples were treated with RQ1 RNase-free DNase (Promega) for 30 min at 37˚ C. RNA was then extracted using TRIzol Reagent (Thermo Fisher Scientific) and resuspended in DEPC-treated H$_2$O. RNA was also extracted from untransfected S2 cells to determine the level of editing in the ADAR$_{CD}$ control, as well as to account for any single nucleotide polymorphisms present in the S2 line.

The integrity of total RNA samples were assessed on a Bioanalyzer 2100 using an RNA 6000 Pico chip (Agilent Technologies, CA). Poly-A containing RNA transcripts were enriched from 2 µg of total RNA for each sample using oligo-dT magnetic beads, and underwent further rRNA-depletion using the RiboZero (Human-Mouse-Rat) kit (Illumina, CA). The final mRNA samples were fragmented and converted to cDNA and barcoded Illumina sequencing libraries were prepared using the PrepX RNA-seq library kit on the automated Apollo 324 NGS Library Prep System (TakaraBio, CA). The libraries were examined on Agilent Bioanalyzer DNA High Sensitivity chips for size distribution, quantified by Qubit fluorometer (Invitrogen, CA), then pooled at equal molar amounts and sequenced on Illumina HiSeq 2500 Rapid flowcells as single-end 180 nt reads. Raw sequencing reads were filtered by Illumina HiSeq Control Software and only the Pass-Filter (PF) reads were used for further analysis. Each sample had approximately 90 million reads. Two biological replicates were generated and analyzed for each condition.

Computational analysis was performed as outlined in Rahman et al. [62] to identify editing events present in Fne-ADAR$_{CD}$ and ADAR$_{CD}$. The thresholds used in Biswas et al. [63] were applied here, namely a minimum of 20 reads and a minimum editing frequency of 5%. As expected, the ADAR$_{CD}$ control yielded very few editing events (100–300 edits), whereas Fne-ADAR$_{CD}$ produced >1200 editing events in each of two replicates. Integrative Genomics Viewer (IGV) was used to view the bedgraph data and to analyze the sequence surrounding the editing sites. The PANTHER GO term overrepresentation test was used to evaluate the biological processes of all 1035 potential targets that appeared in one or both TRIBE replicates. Editing and read count information from a single trial was used to analyze editing percentage as a function of number of reads.

## RNA immunoprecipitation

Expression of Fne-V5 in stably transfected S2 cells was induced with CuSO$_4$ for 24 h and cells were then harvested, resuspended in IP buffer (10 mM Tris pH 7.5, 250 mM NaCl, 0.5mM EDTA, 0.5% IGEPAL), and centrifuged for 25 min at 15000 rpm at 4˚ C. Uninduced control cells were processed in parallel. The lysates were precleared using Protein G agarose beads (Thermo Fisher Scientific) for 30 min at 4˚ C. Beads were incubated with 15 µg anti-V5 antibody for 1 h at RT, then washed with PBS followed by IP buffer. Precleared lysates were incubated with V5-coupled Protein G agarose beads for 1 h at 4˚ C under constant mixing. The beads were then washed, and half of the beads were used for immunoblotting, and the remaining half were used for RNA isolation. To isolate RNA, beads were incubated with RQ1 RNase-free DNase for 30 min at 37˚ C, after which RNA was extracted using TRIzol and resuspended in DEPC-treated H$_2$O. cDNA was generated using SuperScript II reverse transcriptase (Invitrogen) and oligo(dT) primer, according to the manufacturer's protocol. For −RT controls, DEPC-treated H$_2$O was added instead of reverse transcriptase. The primers listed below were used for RT-PCR analysis.

*rngo*: Fw- 5'-AGGTCTTCTGCTTGGACGTG-3'; Rev- 5'-AAACATCTGTCGCACCGTCT-3'

*bsg*: Fw- 5'-ATCACTTGATAAGCTGGTGCC-3'; Rev- 5'-CCATCGTCGTTCGTATCCGT-3'
 *cdc42*: Fw- 5'-TGCAAACCATCAAGTGCGTG-3'; Rev- 5'-GCGTCTTTTGGCAATGGT GT-3'

*pabp*: Fw- 5'-CAGGCTCTCAATGGCAAGGA-3'; Rev- 5'-TGATTTGACGGAAGGGTCGG-3'
 *lost*: Fw- 5'-TCTGGACATCGGCCTCCTTA-3'; Rev- 5'-TGGATTTTCCCGCCTTCTCC-3'

## Statistical analysis

Statistical analysis was performed as described in the figure legends. Two-tailed Student's *t*-test was use for comparison between two groups. To compare three or more groups, a one-way ANOVA with Bonferroni-Holm *post hoc* test was used if the data displayed normal distribution, as confirmed by the Shapiro-Wilk test, and similar variance. For data that did not meet these standards, a Kruskal-Wallis with a Dunn's *post hoc* test was used. For Gene Ontology analysis (Fig 5E), a Fisher's exact test was used. Source data for graphs are provided in S3 Table (main text figures) and S4 Table (supporting figures).

## Supporting information

**S1 Fig. Fne expression in class IV da neurons.** (A-C, A'-C') Confocal z-series projections of wild-type class IV da neurons labeled with anti-Fne (A-C) and anti-RFP to detect the CD4-Tom membrane marker (A'-C'). *ppk-GAL4* was used to drive expression of *UAS-CD4-td-Tom*. The class IV da neuron in each set of images is circled. Other classes of da neurons visible in (A) are indicated with white arrowheads, yellow arrowheads indicate external sensory (ES) neurons. (D) Quantification of Fne levels in class IV da neurons at L2 (n = 12 neurons), L3 (n = 14 neurons), and 10 h after puparium formation (APF; n = 5 neurons). Values are mean ± s.e.m.; **p<0.01 as determined by one-way ANOVA with Bonferroni-Holm *post hoc* test. Scale bar: 20 μm.
(TIF)

**S2 Fig. Fne acts cell autonomously in class IV da neurons.** (A-C) Confocal z-series projections of wild-type (WT; A), *fne⁻* (B) and *fne⁻; UAS-fne* (C) neurons. *ppk-GAL4* was used to express *UAS-CD4-tdTom* to mark the neurons and to drive *UAS-fne* expression. (D) Quantification of empty space in the arbor for WT (n = 11 neurons), *fne⁻* (n = 12 neurons), and *fne⁻; UAS-fne* (n = 10 neurons). (E-G) Confocal z-series projections of WT (E), *fneRNAi* (VDRC) (F) and *fneRNAi* (TRiP) (G) neurons. *ppk-GAL4* was used to express *UAS-CD4-tdTom* to mark the neurons and to drive *UAS-fneRNAi* expression. Enlargements of the boxed areas in (E-G) are shown in E'-G'. (H) Quantification of empty space in the arbor for WT (n = 12 neurons), *fneRNAi* (VDRC) (n = 14 neurons), and *fneRNAi* (TRiP) (n = 12 neurons). (I, J) Confocal z-series projections of WT (I) and *fneRNAi* (VDRC) (J) neurons. Pan-epidermal expression of *UAS-fneRNAiVDRC* was driven using *GAL469B* and class IV da neurons were marked using *ppk-CD4-tdGFP*. (K) Quantification of empty space in the arbor for WT (n = 12 neurons) and *fneRNAi* (VDRC) (n = 13 neurons). (L, M) Confocal z-series projections of WT (L) and *fneRNAi* (VDRC) (M) neurons. Pan-epidermal expression of *UAS-fneRNAiVDRC* was driven using *GAL458A* and class IV da neurons were marked using *ppk-CD-tdTom*. (N) Quantification of empty space in the arbor for WT (n = 11 neurons) and *fneRNAi* (VDRC) (n = 10 neurons). Values are mean ± s.e.m.; **p<0.01 as determined by one-way ANOVA with Bonferroni-Holm *post hoc* test (D, H); ns = not significant as determined by two-tailed Student's *t*-test (K, N).

Scale bars: 50 μm (A, E, I, L), 20 μm (E').
(TIF)

**S3 Fig. Fne acts as a brake on pruning of class IV da neurons.** (A, B) Confocal z-series projections of wild-type (WT; A) and *fne⁻* (B) neurons 7 h APF. (C, D) Confocal z-series projections of WT (C) and *fne^OE* (D) neurons at 16 h APF. *ppk-GAL4* was used to drive expression of *UAS-fne*. For all genotypes, *ppk-GAL4* was used to express *UAS-CD4-tdGFP* to mark the neurons. (E) Quantification of primary and secondary dendrites attached to the soma at 7 h AFP and 16 h APF: n = 8 neurons (WT, 7 h APF), 12 neurons (*fne⁻*, 7 h APF), 8 neurons (WT, 16 h APF), 6 neurons (*fne^OE*, 16 h APF). Values are mean ± s.e.m.; ***p<0.001 as determined by two-tailed Student's *t*-test. Scale bars: (A) 50 μm; (C) 100 μm.
(TIF)

**S4 Fig. Changes in acetylated tubulin are similar to those observed with Futsch in *fne* mutant and overexpressing neurons.** (A-C) Confocal z-series projections of wild-type (WT; A), *fne⁻* (B) and *fne^OE* (C) neurons. *ppk-GAL4* was used to express *UAS-CD4-tdTom* to mark the neurons and to express *UAS-fne*. Images show anti-acetylated tubulin (magenta, A-C, A'-C') to label stable microtubules and anti-RFP (green; A'-C') to detect CD4-Tom. Arrows indicate branches positive for acetylated tubulin. (D) Quantification of acetylated tubulin intensity normalized to the intensity of the membrane marker for WT (n = 9 neurons), *fne⁻* (n = 10 neurons), and *fne^OE* (n = 13 neurons). Values are mean ± s.e.m. One-way ANOVA with Bonferroni-Holm *post hoc* test indicated no significant difference between the genotypes. Scale bar: 50 μm.
(TIF)

**S5 Fig. Ectopic expression of *fne* in *Drosophila* ovarian follicle cells leads to changes in integrin and cell shape.** (A, A', B, B') Confocal z-series projections showing anti-Mys immunofluorescence (red) in wild-type follicle cells (WT; A. A') or follicle cells expressing *UAS-fne* driven by *tj-GAL4* (B, B'). Follicle cells (fc), nurse cells (nc) and the oocyte (oo) are labeled. (C, C', D, D') Enlargements of the indicated areas in (A' and B'). Nuclei are labeled with DAPI (blue) in (A'-D'). (E) Quantification of anti-Mys immunofluorescence: n = 19 follicle cells (WT), 16 follicle cells (*fne^OE*) from at least four egg chambers. Values are mean ± s.e.m.; ***p<0.001 as determined by two-tailed Student's *t*-test. Scale bar: 10 μm.
(TIF)

**S6 Fig. Expression of TRIBE proteins in *Drosophila* S2 cells.** (A) Schematic of ADAR_CD and Fne-ADAR_CD constructs that were expressed in S2 cells. pMT denotes the metallothionine promoter and V5 denotes the V5 epitope tag. (B) Western blot of protein isolated from stable cell lines before and after induction with CuSO₄. Anti-V5 antibody was used to detect ADAR_CD and Fne-ADAR_CD, anti-Khc antibody was used to detect Kinesin heavy chain (Khc) as a loading control.
(TIF)

**S7 Fig. Fne-V5 binds transcripts identified in TRIBE.** (A) Anti-V5 immunoprecipitation of cell lysates obtained from stable cell lines with or without induction with CuSO₄. Anti-V5 antibody was used to detect Fne-V5. (B) Whole cell extract and immunoprecipitates were analyzed by RT-PCR to detect *bsg*, *cdc42*, *pabp and lost*. *rngo* was used as a negative control, as it is abundant in *Drosophila* S2 cells, but was not identified as a target of Fne by TRIBE and does not show overlap with the enriched GO terms for the set of target RNAs.
(TIF)

**S8 Fig. Independent validation of *bsg* RNA*i* phenotype.** (A–D) Confocal z-series projections of wild-type (WT; A), *bsg*$^{RNAi}$ (B), *fne*$^−$ (C) and *fne*$^−$; *bsg*$^{RNAi}$ (D) neurons. *ppk-GAL4* was used to drive expression of *UAS-CD4-tdTom* to mark the neurons and to drive expression of *UAS-bsgRNAi* (BDSC 52110). (E) Quantification of empty space in the arbor for WT (n = 9 neurons), *bsg*$^{RNAi}$ (n = 7 neurons), *fne*$^−$ (n = 6 neurons), and *fne*$^−$; *UAS-bsg*$^{RNAi}$ (n = 7 neurons). (F, G) Confocal z-series projection of neurons from a WT (F) larva and a larva expressing *UAS-bsgRNAi* (BDSC 52110) in the posterior portion of the epidermis driven by *hh-GAL4* (G). Neurons were marked with *ppk-CD4-tdGFP*. (H) Quantification of empty space in the posterior third of the arbor for WT (n = 17 neurons) and *bsg*$^{RNAi}$ (n = 16 neurons) in (F, G). Values are mean ± s.e.m.; $^{**}$p<0.01 as determined by one-way ANOVA with Bonferroni-Holm *post hoc* test; ns = not significant. Scale bars: 50 μm.
(TIF)

**S9 Fig. Bsg localizes to enclosed dendrite segments.** Anti-HRP immunofluorescence (red) of a late L3 larva expressing Bsg::GFP (white). Anti-HRP immunostaining was performed prior to tissue permeabilization and therefore detects unenclosed dendrite segments (see Materials and methods). Bsg::GFP localizes to areas of dendrite enclosure, as marked by the absence of HRP signal (yellow arrows in merged image). Note that HRP detects other unenclosed neurons in the field. Scale bar: 50 μm.
(TIF)

**S1 Table. Editing sites identified using TRIBE.** Edited sites identified in S2 cells expressing the ADAR$_{CD}$ control or Fne-ADAR. All edited sites listed have met the thresholds of 5% editing and 20 reads. Information is provided in bedgraph format and includes gene name, number of edited sites, editing frequency, and the chromosomal location of edited sites.
(XLSX)

**S2 Table. Targets of Fne.** List of genes with sites edited in both Fne-ADAR biological replicates and the chromosomal coordinates of the edits in common between the two replicates. These edited sites meet the thresholds of 5% editing and 20 reads and are not present in the ADAR$_{CD}$ control.
(XLSX)

**S3 Table. Source data for graphs in main text figures.**
(XLSX)

**S4 Table. Source data for graphs in supporting figures.**
(XLSX)

## Acknowledgments

We are grateful M.L. Samson for fly stocks and the Fne antibody, A. Ephrussi, K. Broadie, and J. Parrish for fly stocks, and M. Soller for plasmid DNA. We thank the G. Laevsky and the Princeton Confocal Imaging Facility for assistance with confocal imaging, the Princeton Genomics Core Facility for library preparation and RNA-seq analysis, R. Leach for assistance with the TRIBE analysis, and H. Li, A. Hakes and J. Tamayo for comments on the manuscript.

## Author Contributions

**Conceptualization:** Rebecca A. Alizzi, Conrad M. Tenenbaum, Elizabeth R. Gavis.

**Formal analysis:** Rebecca A. Alizzi, Derek Xu.

**Funding acquisition:** Elizabeth R. Gavis.

**Investigation:** Rebecca A. Alizzi, Derek Xu.

**Methodology:** Wei Wang.

**Project administration:** Elizabeth R. Gavis.

**Supervision:** Elizabeth R. Gavis.

**Writing – original draft:** Rebecca A. Alizzi, Elizabeth R. Gavis.

**Writing – review & editing:** Derek Xu, Conrad M. Tenenbaum.

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
