## [Decision Letter · Decision Letter 0]

12 Jul 2020

Dear Dr. Gavis,

Thank you very much for submitting your Research Article entitled 'The ELAV/Hu protein Found in neurons regulates cytoskeletal and ECM adhesion inputs for space-filling dendrite growth' to PLOS Genetics. Your manuscript was fully evaluated at the editorial level and by independent peer reviewers. The reviewers appreciated the attention to an important problem, but raised some substantial concerns about the current manuscript. Overall, the reviewers felt that the characterization of a molecular mechanism by which an RNA-binding protein (Fne) would control dendrite morphogenesis would be of interest, but they expressed concerns regarding the support for the proposed relationship between Fne and Bsg (in particular whether the effects on dendrite development are due to Fne acting in neurons or epithelial cells) and the TRIBE analysis (see specific comments from Reviewer #2). A recent report on Bsg by Hunter et al. 2020 (PMID 32265259) pointed out by Reviewer #2 is likely relevant to this study. Based on the reviews, we will not be able to accept this version of the manuscript, but we would be willing to review again a much-revised version. We cannot, of course, promise publication at that time.

If you decide to revise the manuscript for further consideration at PLOS Genetics, please aim to resubmit within the next 60 days, unless it will take extra time to address the concerns of the reviewers, in which case we would appreciate an expected resubmission date by email to plosgenetics@plos.org.

[LINK]

We are sorry that we cannot be more positive about your manuscript at this stage. Please do not hesitate to contact us if you have any concerns or questions.

Yours sincerely,

Jill Wildonger

Guest Editor

PLOS Genetics

Gregory Barsh

Editor-in-Chief

PLOS Genetics

Reviewer's Responses to Questions

**Comments to the Authors:**

Reviewer #1: The manuscript by Alizzi et al describe the characterization of Fne (ELAV/Hu) functions in class IV da neurons of Drosophila. The fne mutant was identified in their previous screening for RNA binding proteins in dendrite development. In continuation, they show that fne mutants display unstable dendritic branches (and enclosure and pruning phenotypes). They found that Futsch staining for stable microtubules was reduced. In a search for target RNAs bound by Fne, they identified bsg as a target and analysis of bsg mutant suggest bsg mediates some of fne mutant phenotype such as the shorter and extra dendrites and lack of microtubule stability.

The molecular mechanism for Fne proposed in the manuscript is the most interesting part of the manuscript. Yet the phenotypic characterization is not complete and the cell biology study could be further improved. Genetic study also needs strengthened to substantiate the conclusion. Please see following comments/concerns.

Major concerns:

1. The fne mutant displays quite a few interesting stage-dependent dendritic phenotypes, such as extra proximal and shorter branches in the L2 stage, raising the possibility that Fne’s function or expression is temporally (and spatially) regulated. There is only one image showing Fne expression in class IV da neurons (Fig. S1), and the specific larval stage for image was not described. What are Fne expressions in different larval and pupal stages? What are the cells/neurons that are also Fne-positive in Fig S1 (in addition to the class IV da neuron)?

2. If Fne is also expressed in other classes of da neurons, do they show dendrite phenotypes in fne mutants? In particular the class I da neuron which is not space-filling but microtubule-dominant in dendrites, a contrast to the class IV da neurons.

3. The cytoplasmic distribution of Fne in Fig S1 does not correlate with local action of Fne to regulate microtubules and Bsg in dendrites. Have fluorescence protein-Fne fusions been examined in the dendrites to examine the protein localization?

4. Microtubule phenotype in fne mutants is suggested only by the reduction of Futsch immunostaining. Other microtubule makers should be examined to suggest the reduction in stable microtubules.

5. The role of Fne in the integrin pathway for dendrite-ECM interaction is very indirect, as suggested by the increases in branch enclosure and cross-over. One would expect to perform genetic interaction with integrin subunits to suggest an involvement. The overexpression of Fne to disturb Mys localization in follicle epithelial cells is not supportive, either. What is the localization of Myc in loss of fne mutants?

6. The authors describe that “Epidermal knockdown of bsg using hh-GAL4 led to similar defects” to ppk-GAL4 knockdown. This is not clear! The hh-GAL4 knockdown only affect the posterior portion of the segment. Does the dendrite phenotype is also present in the anterior part?

7. Since Bsg also regulates microtubule stability, what are the terminal branch dynamics in bsg mutants? Does it phenocopy fne mutant in branch dynamics?

Minor comments

1. The usage of “CD4:GFP” and “CD4:TOM” are not conventional! Please check the flybase and correct them!

2. Fig 1I showing the Sholl analysis is not very clear! It could be separated into two sub-figures for L2 and L3.

3. In Fig 1L, the number of crossovers per 1000um were scored. It is not clear what are does the “1000um” stand for ( not described in Fig legend!)? One way is to score crossover per branch!

4. What are the terminal phenotypes in pruning? Do dendrites regrew in fne mutants after premature pruning?

5. There are two Fig 5B’s in Fig 5!

Reviewer #2: Review is uploaded as an attachment.

Reviewer #3: The manuscript by Alizzi et al. describes the role of the RNA-binding protein Fne in dendrite morphogenesis of Drosophila da neurons. The authors identified defects in microtubule stabilization and dendrite-substrate attachment, which resulted in altered dendritic dynamics and reduced dendrite coverage. The authors then further searched for Fne target RNAs using the TRIBE technology and identified Bsg as one target that is responsible for at least partial function of Fne. Overall, the findings in this study suggest new mechanisms of how dendrite development is regulated by RNA-binding proteins and will be valuable to the field if the conclusions are well supported. However, I also have major concerns about the quality of some of the evidence for supporting their conclusions and the rigor of some their experiments. I think these issues need to be addressed before the manuscript can be considered for publication.

Major concerns:

1. A very important conclusion regarding the role of Fne is that it functions cell-autonomously to regulate neuronal cytoskeleton and dendrite-substrate interactions. However, in my opinion, this cell-autonomy has not been well established in the manuscript. I understand that a previous study suggests that Fne is specific to neurons based on in situ data and antibody staining. However, these expression analyses may not be sensitive enough to detect low expression in other tissues. In this regard, it is premature to imply that only neurons are affected in the whole animal mutant (e.g. referring to neurons in fne mutant as fne mutant neurons). As this is a very important foundation of this paper, can the authors show other evidence that Fne does not have a role in tissues like epidermal cells (e.g. RNAi, MARCM, etc). Related to this point, does the image of Fne staining in Figure S1 include epidermal sections?

2. Related to the first comment, Bsg::GFP expression levels/patterns on epidermal cells (outside dendrites) seem to be altered in the fne mutant (Figure 7). Can this be due to the epidermal function of Fne? Can the authors provide other evidence that neuronal Fne affects recruitment of epidermal Bsg::GFP to dendrites? A few ideas include overexpressing untagged Bsg::GFP in neurons, overexpressing Fne in neurons in the WT, and overexpressing Fne in the fne mutant.

3. The authors concluded that “Fne positively regulates Bsg levels in the neuron” (page 12). However, direct evidence is absent. Can Fne affect Bsg distribution instead of expression level? For example, in S2 cells expressing Fne, is Bsg mRNA level increased? I also don’t think the provided data convincingly show that Bsg tracks are derived from dendrites (Figure 6C”). In RNAi cells, there are still residual levels of Bsg::GFP in epidermal cells. The tracks could be epidermal Bsg proteins enriched at dendrite enclosure sites, like many other cell junctional proteins. A support for this argument is that Bsg::GFP tracks along dendrites are much brighter on the WT control epidermal cells.

4. The results of Futsch staining shown in the manuscript are confusing and directly contradict many published results in the literature. For example, Futsch staining should be detected in dendrites of all da neurons, but Figures 3 and 7 suggest that it is specific to class IV da neurons. Second, the staining signal should not be limited to only the main trunks of primary dendrites but should also be in other stabilized dendrites even in the distal arbor. My assumption is that the limited distribution in neurons shown in these figures is due to low resolution and/or low detection power. Can the authors show higher-rez images with more sensitive detection?

5. I think the integrin expression data shown in Figure 4 are not very relevant. What does the change of integrin level/distribution in follicle cells tell us about the regulation in neurons? Also the results were not well described or well interpreted. Can the changes be simply in protein localization instead of expression (there seems to be more intracellular Mys in fne mutant). Are the changes in cell shape caused by changes of integrin?

6. The manuscript is generally lacking in statistical rigor, at least in the current form. First, the exact sample size for each experiment was not provided. In addition, some experiments had inadequate sample sizes (Figure 2D-E, Figure S2, and Figure 2H). In general, the sample size should be larger than 10. Lastly, wrong statistical methods were used for some experiments. For example, one-way ANOVA instead of t-test should be used when more than two groups are being compared (Figure 1E-1J, 2D-2E, 3D, 7G-7L), assuming that the data meet the criteria (normal distribution, equal variances, etc) for one-way ANOVA.

7. I am not very sure if the real-time (should be called time-lapse, to be technically correct) imaging provided sufficiently convincing data for drawing a conclusion. First, the time duration seems a little too short to clearly visualize the growth dynamics. Second, the sample size is too low. Regarding the analysis, did the authors consider the change of dendrite length? Are the distances of dendrites to the soma considered, since fne LOF affects distal and proximal dendrites differently?

Minor concerns:

1. Can the authors add some discussion to compare the roles of Fne and vertebrate HuD in dendrite morphogenesis?

2. The fneOE data shown in Figure 1I and 1J seem inconsistent. fneOE has a larger critical radius than fne in 1J, but it seems to have a smaller critical radius in 1I.

3. The fne and fneOE phenotypes in pruning are very interesting, but the text has very little interpretation or discussion about these results. It would be good to discuss it in the discussion.

4. There is very little explanation about the assay for detecting dendrite enclosure. I understand that the assay has been published. But some explanation is needed for readers to understand this experiment.

5. The descriptions for some results are vague. For example, Bsg is said to be expressed around the periphery of epidermal cells” (page 11). Does “periphery” mean the cell border, the plasma membrane, or something else?

6. Please indicate RNAi-expressing cells in hh-Gal4 RNAi images.

7. It’s not clear from the text, figure legends, or methods how some of the image panels were created (Figure 7A’, B’).

8. The data about the effects of bsg RNAi on dendrite crossing were not shown.

**Have all data underlying the figures and results presented in the manuscript been provided?**

Reviewer #1: **No: **spread sheets could be provided for bar graphs, Fig 1E-J, L; Fig 2D, E, H, Fig 3D, Fig 4 B, C, H, Fig 5B-D; Fig 6 F, I, Fig 7G, L and those in supplemental figures.

Reviewer #2: Yes

Reviewer #3: None

PLOS authors have the option to publish the peer review history of their article (what does this mean?). If published, this will include your full peer review and any attached files.

Reviewer #1: No

Reviewer #2: No

Reviewer #3: No

---

## [Editor Report · Decision Letter 1]

29 Oct 2020

Dear Dr Gavis,

We are pleased to inform you that your manuscript entitled "The ELAV/Hu protein Found in neurons regulates cytoskeletal and ECM adhesion inputs for space-filling dendrite growth" has been editorially accepted for publication in PLOS Genetics. Congratulations!

Yours sincerely,

Jill Wildonger

Guest Editor

PLOS Genetics

Gregory Barsh

Editor-in-Chief

PLOS Genetics

Comments from the reviewers (if applicable):

**Data Deposition**

http://datadryad.org/submit?journalID=pgenetics&manu=PGENETICS-D-20-00942R1

**Press Queries**

---

## [Editor Report · Acceptance letter]

9 Dec 2020

PGENETICS-D-20-00942R1 

The ELAV/Hu protein Found in neurons regulates cytoskeletal and ECM adhesion inputs for space-filling dendrite growth 

Dear Dr Gavis, 

We are pleased to inform you that your manuscript entitled "The ELAV/Hu protein Found in neurons regulates cytoskeletal and ECM adhesion inputs for space-filling dendrite growth" has been formally accepted for publication in PLOS Genetics! Your manuscript is now with our production department and you will be notified of the publication date in due course.

With kind regards,

Nicola Davies

PLOS Genetics

On behalf of:
